# A multi-modal, asymmetric, weighted, and signed description of anatomical connectivity

Jacob Tanner[1,2], Joshua Faskowitz[3], Andreia Sofia Teixeira[4], Caio Seguin[3], Ludovico Coletta[5], Alessandro Gozzi[6], Bratislav Mišić[7] & Richard F. Betzel[1,2,3,8] ✉

The macroscale connectome is the network of physical, white-matter tracts between brain areas. The connections are generally weighted and their values interpreted as measures of communication efficacy. In most applications, weights are either assigned based on imaging features–e.g. diffusion parameters–or inferred using statistical models. In reality, the ground-truth weights are unknown, motivating the exploration of alternative edge weighting schemes. Here, we explore a multi-modal, regression-based model that endows reconstructed fiber tracts with directed and signed weights. We find that the model fits observed data well, outperforming a suite of null models. The estimated weights are subject-specific and highly reliable, even when fit using relatively few training samples, and the networks maintain a number of desirable features. In summary, we offer a simple framework for weighting connectome data, demonstrating both its ease of implementation while benchmarking its utility for typical connectome analyses, including graph theoretic modeling and brain-behavior associations.

The connectome is an example of an "anatomical" or "structural" network, in that the edges all represent physical, material pathways[1–4]. In anatomical networks, connections are usually associated with weights. In human tractography data, these weights are frequently assigned based on diffusion parameters – e.g. fractional anisotropy, mean diffusivity, or streamline counts – and are interpreted as measures of white-matter fiber integrity.

The connectome is of great interest to a number of scientific communities. Cognitive processes are supported by distributed, brain-wide networks[5,6] and many neuropsychiatric disorders are thought to be disorders of dysconnectivity[7,8]. Mapping connectomes and understanding their organizing and operational principles[9–13] is also a key aim of network neuroscience[14] – the nascent discipline that focuses on

modeling and analyzing brain data (micrographs, MR images, electrophysiological recordings) as networks.

How to assign weights to connectome edges represents an ongoing challenge in network neuroscience. Most studies define edge weights based on microstructural properties and tractographical parameters of the reconstructed fibers–e.g. streamline count (normalized or raw), fractional anisotropy, and mean diffusivity, among others[15–18]. Although this approach has proven eminently fruitful, it nonetheless has limitations. For instance, tractography algorithms can not infer directionality, and most weighting schemes are, therefore, bidirectional and incapable of resolving differences in weights of incoming vs outgoing connections.

[1]Cognitive Science Program, Indiana University, Bloomington, IN, USA. [2]School of Informatics, Computing, and Engineering, Indiana University, Bloomington, IN, USA. [3]Department of Psychological and Brain Sciences, Indiana University, Bloomington, IN, USA. [4]LASIGE, Departamento de Informática, Faculdade de Ciências, Universidade de Lisboa, Lisboa, Portugal. [5]Fondazione Bruno Kessler, Trento, Italy. [6]Functional Neuroimaging Lab, Istituto Italiano di Tecnologia, Center for Neuroscience and Cognitive Systems, Rovereto, Italy. [7]McConnell Brain Imaging Centre, Montréal Neurological Institute, McGill University, Montréal, Canada. [8]Program in Neuroscience, Indiana University, Bloomington, IN, USA. ✉e-mail: rbetzel@indiana.edu

More saliently, weights derived from microstructural properties or diffusion parameters do not incorporate information about brain function. That is, they are derived exclusively from structural and diffusion imaging data, overlooking features derived from functional imaging data–e.g. blood-oxygen-level-dependent signal obtained from functional MRI–that could be used to inform the valence and magnitude of connection weights. Although many studies have sought to relate independently constructed structural and functional networks with one another[19–23], few have examined strategies for incorporating multiple imaging modalities to jointly construct a single network that embodies the most useful aspects of each.

Here, we present a multi-modal, explanatory model for estimating the weights of structural connections. In the spirit of diffusion-/tractography-based models, ours preserves the brain's sparse white-matter architecture. However, rather than assign structural weights based on diffusion/imaging parameters, we assign weights based on the parameters of multi-linear regression models. These models are fit independently for each region, $i$, and predict that region's future activity based on the weighted histories of its connected neighbors. This allows us to fit asymmetric and signed edge weights for networks of hundreds of nodes in a matter of seconds.

Our manuscript aims to explore this model and its network properties, positioning it as an intermediate method, situated between tractography-based weighting schemes and correlation networks derived from functional imaging data. To this end, we find that the model predicts fMRI BOLD activity at a rate greater than chance even when using a relatively small amount of data (approximately 1% of samples). We show that these models exhibit subject specificity and the inferred edge weights are aligned, broadly, with known functional systems, despite the fact that edge weights exhibit imperfect alignments with interregional correlations (see Fig. S1). Further, we show that this network exhibits bilaterally symmetric, hemisphere-spanning communities and a shortest path structure that involves most edges (in contrast with streamline-weighted networks that use only 15% edges in its shortest paths backbone). Taking advantage of the directed nature of links, we find evidence of robust asymmetries in connection weights and regions' connectivity profiles (incoming vs outgoing connections). Finally, in two applications we show that the inferred edge weights systematically reconfigure during movie-watching and across the human lifespan. Collectively, these observations suggest that this model is a practical alternative to existing edge-weighting schemes, and effectively endows anatomical connections with functional properties, thereby opening up avenues for future exploration and applications.

## Results

### Fitting and benchmarking asymmetric, weighted, and signed structural connectivity

Brain regions are linked to one another via white-matter fiber tracts. The topology and edge weights of this network constrain interareal communication and shape patterns of spontaneous activity. Most studies set the weights of structural connections equal to microstructural properties estimated from diffusion weighted images and tractography–e.g. fractional anisotropy or streamline derivatives. However, the ground truth connection weights are unknown, motivating the exploration and benchmarking of alternative weighting schemes.

Here, we use a model-based framework for assigning weights to existing structural connections. Briefly, we follow existing modeling work[24–27] and assume that at time $t$ the state of region $i$ (level of fMRI BOLD activity) is a function of its neighbors' states at time $t - 1$ plus an offset (bias). That is:

$$y_i(t) = \sum_{j \in \Gamma_i, j \neq i} W_{ji} y_j(t-1) + c_i, \tag{1}$$

where $\Gamma_i$ is the set of region $i$'s connected neighbors. We use linear regression and ordinary least squares to estimate the parameters $W_{ji}$ and $c_i$ separately for each node $i$ (Fig. 1a, b). Thus, the resulting matrix $W \in \mathbb{R}^{n \times n}$, is sparse and preserves exactly the binary structure of white-matter connectivity (Fig. 1c). However, the weights, which represent regression coefficients, can take on either positive or negative valence, whereas weights are typically positive only for connectomes inferred from dMRI and tractography. We note also that this network is directed–i.e. in general, $W_{ij} \neq W_{ji}$.

In this section, we report the results of the fitted model on resting-state fMRI data from human subjects. That is, we describe basic features of this asymmetric, weighted, and signed connectome and contrast them with a connectome in which weights are defined using a commonly used metric–i.e. streamline density (streamline count divided by geometric mean of regional volumes)[10,28].

We fit the model at the group level using pooled time series data from 95 participants from the Human Connectome Project[29] (HCP100UR, five subjects excluded due to incomplete data or quality issues) and a group-averaged binary SC matrix[30]. We found that, at the group level, the model performed well (correlation between observed and predicted activity from individual scans, $r = 0.76 \pm 0.03$; mean squared error, $MSE = 0.43 \pm 0.05$; Fig. 1d, e). We also found that the model weights stabilize with relatively few samples. Specifically, we randomly sub-sampled an equal number of frames from each participant and scan and used those frames to estimate the connection (regression) weights. We repeated this process 100 times while varying the number of samples from 2, 4, 8, 16, 32, 64, 128, 256, 512, to 1099. We found that even with approximately 6% of the total number of samples (64 samples per scan), the estimated weights achieved a correlation with the full-sample weights of $r = 0.993$; Fig. 1f).

Brain activity dynamics and its correlation structure are deeply individualized[31,32]. A good model of brain activity, therefore, must also exhibit subject specificity. To assess whether model performance was, indeed, subject-specific, we estimated weights using three of every subject's four resting state scans, and used those weights to predict the activity of the held-out scan (as well as the activity of all other scans and subjects; Fig. 1g). We found that the error (mean squared error) was lower for the held-out scans than for the scans of any other subjects (two-sample $t$-test; $p < 10^{-15}$; Fig. 1h). Here, as in subsequent single-subject-/-scan analyses, we fit edge weights using the same group-representative connectivity mask. This ensures that any differences between individuals are not driven by differences in the underlying anatomical connectivity, but driven jointly by differences in edge weights and resting brain dynamics. In a supplementary analysis, we also show that the weights of models with subject-specific fMRI as well as subject-specific structural connectivity are more similar within subjects, than between subjects (Fig. S20a, b; two-sample $t$-test; $p < 10^{-15}$). Importantly, we also found that the subject-specific model weights were highly similar to the group-estimated model weights (Fig. S20c; mean similarity $r = 0.66$; one-sample $t$-test; $p < 10^{-15}$).

Next, we assessed whether the observed results, namely the model error, was consistent with chance. Accordingly, we compared the observed model fitness against null distributions obtained under five distinct null models[33](see "Materials and methods" section for details related to these null models). In general, we found that the error (MSE) was significantly lower using the intact data than in any of the null models (two-sample $t$-test, $p < 10^{-15}$; Fig. 1i).

Finally, we examined some of the basic properties of the weights fit at the group level. We found that both positive and negative connection weights decay monotonically with distance (Fig. 1j). However, for any given distance bin there was a range of edge weight values. Examining the most extreme (z-scored weight of $z > 3$ relative to the other edges in the same bin), we find they are dominated by intra-hemispheric connections ($\approx$71%). Although fewer in number, the

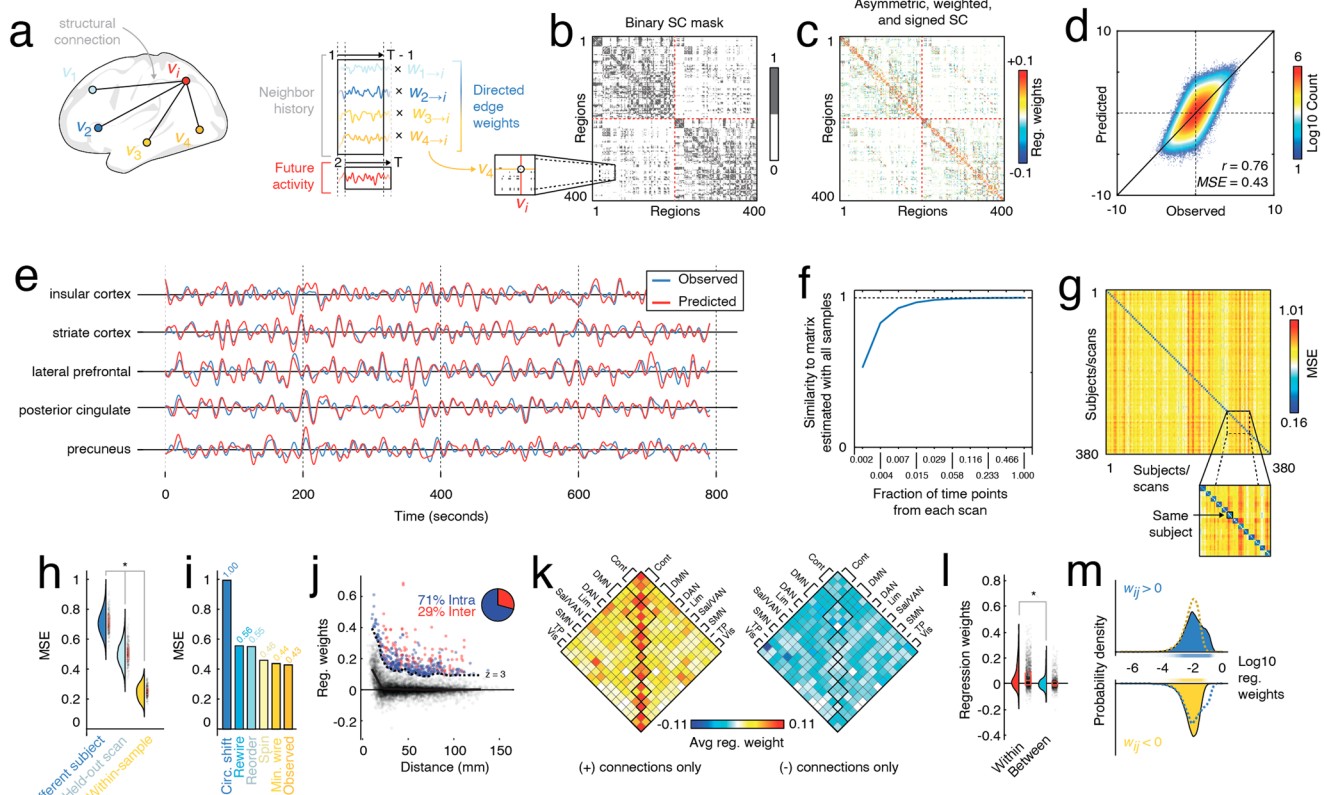

**Fig. 1 | Fitting and characterizing weighted, signed, and asymmetric structural connectivity. a** Here, we used linear model to estimate regression (edge) weights. For a given region $i$, we identified its structurally connected neighbors and used their past activity to predict node $i$'s future activity. This procedure results in a series of regression weights; one weight is associated with each neighbor. **b** Those weights were then entered into the binary structural connectivity "mask". When this procedure was repeated for all $i \in [1, ..., n]$ it generates a weighted, signed, and asymmetric matrix whose nonzero entries are masked by structural connections (see panel **c**). **d** Two-dimensional histogram showing observed and predicted activity, pooled across all participants and scans. Colors indicate number of samples in any bin. **e** Examples of observed and predicted activity for five select regions in a single subject and scan. **f** Similarity of regression weights (edge weights) as a function of amount of data. Note that units on $x$-axis are expressed as fraction of time points in scan, where the total number of frames was 1099. **g** Weights fit using scans from subject $s$ are better at predicting activity in held-out scans from $s$ than other subjects, $s'$. In this figure, the blocks along the diagonal are 4 × 4 and represent the four resting-state HCP scans. **h** Distributions of errors in model fit, grouped by whether the model is being used to predict activity in a scan of a different subject ($N = 142880$), the held-out scan from the same subject ($N = 1140$), or one of the scans used in fitting the weights ($N = 380$). **i** Comparison of model errors using the observed network with a minimally wired network, one in which rows/columns were randomly reordered, and another in which time series were circularly shifted (independently across regions, scans, and subjects). **j** Relationship of regression weights and Euclidean distance. We also identified edges whose regression weights were much stronger than expected (those above the dashed line). **k** Distribution of regression weights across canonical brain systems (Cont = Control, DMN = default mode network, DAN = dorsal attention network, Lim = limbic, Sal/VAN = salience, SMN = somatomotor, TP = temporoparietal, Vis = visual). **l** Comparison of within-($N = 4654$) and between-system ($N = 24,370$) regression weights. All box plots, shown in red and overlaid on data points in **e** and **f**, depict the interquartile range (IQR) and the median value of the distribution. Whiskers extend to the nearest points ± 1.5 × IQR above and below the 25th and 75th percentiles. Asterisks indicate a significant difference between box plots. **m** Comparison of positive and negative weights.

remaining 29% of connections still exceeds the baseline rate of interhemispheric connections (19.5%).

Next, we tested how positive and negative connections were distributed with respect to canonical brain systems[34]. We found that, within systems, connections tended to be strong and positive whereas negatively-weighted connections showed no clear preference for falling either within or between systems. Indeed, when we examine the weights of individual connections, rather than system averages, we still find that within-system weights tend to be stronger and more positive compared to between-system weights (two-sample $t$-test, $p < 10^{-15}$; Fig. 1l) and that, in general, the mean positive connection is greater than the absolute mean negative (two-sample $t$-test, $p < 10^{-15}$; Fig. 1m).

In the supplementary material we perform several additional tests. These include assessing model performance at different lags (Fig. S2), assessing the relative contributions of long vs short connections (Fig. S3), comparing the estimated edge weights with other measures of functional and structural connectivity (Fig. S1), assessing regional fitness (Fig. S4), assessing the impact of global signal

regression on results (Fig. S5), confirming that the distance dependence of edge weights is preserved when we use curvilinear fiber length rather than Euclidean distance (Fig. S6), fitting edge weights with regularization (Fig. S7), and comparing the relative performance of the asymmetric, weighted, and signed matrix vs the fiber density matrix as structural constraints for dynamic, neural mass models (Fig. S8).

In addition, we performed several analyses to assess how our model is effected by changes to the underlying structural network. We found that as existing structural connections are replaced with non-existing connections (with equal distance) performance of the model degrades (Fig. S16). Furthermore, we found that, in general, changes to the underlying structural network – for example by changing parameters in the estimation of group consensus structural connectivity – result in changes to the weights of the model. More specifically, the similarity of model weights is positively related to the similarity of the structural network (Figs. S18 and S19).). Finally, although the weights of these models do change with changes to the underlying structural

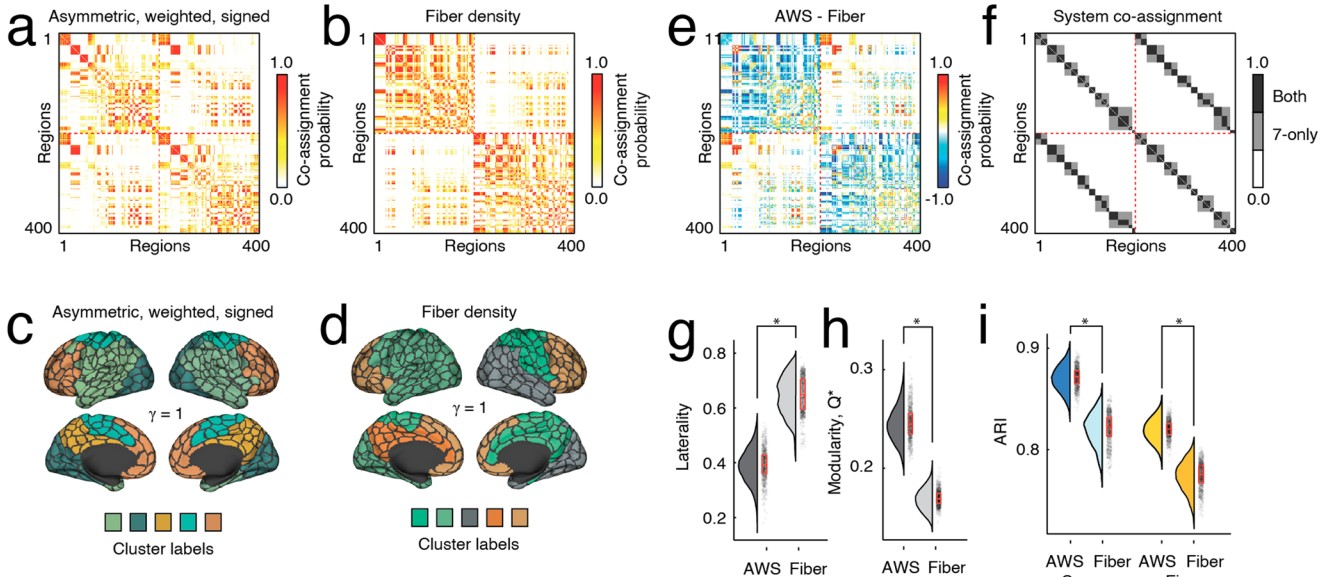

**Fig. 2 | Comparing modular structure of structural networks.** Modules are cohesive subnetworks – nodes that make more connections to other members of the same module than to other modules. Here, we compare the modular structure of network with original weights and the same network with weighted, directed, and signed edges. Here, we examine modules estimated with a fixed resolution parameter ($\gamma = 1$) but explore the multiscale modular structure in the supplement. Co-assignment probability matrices for the inferred edge weight (**a**) and the fiber density matrices (**b**). **c, d** Consensus communities for both versions of weights. **e** Element-wise difference in module co-assignment. **f** System co-assignment matrix for reference. Black entries refer to pairs of brain regions that are assigned to the same system in both coarse and fine-scale system divisions. Gray entries are co-assigned to the same system for the coarse division, only. Comparison of laterality (**g**) and modularity (**h**) of detected modules ($N = 1000$ for each box plot). **i** Alignment of modules with respect to coarse and fine-scale system partitions. Each point in panels **g**–**i** represents a partition from one of 1000 runs of the Louvain algorithm for optimizing modularity. Box plots, shown in red and overlaid on data points in **e** and **f**, depict the interquartile range (IQR) and the median value of the distribution. Whiskers extend to the nearest points ± 1.5 × IQR above and below the 25th and 75th percentiles. Asterisks indicate a significant difference between box plots ($N = 1000$ for each box plot).

network, the lower bound on weight similarity is still reasonably high (between $r = 0.7$ and $r = 0.85$; see Figs. S19e and S18).

In summary, we show that this simple regression framework reliably estimates structural connection weights and requires relatively few observations to do so. The inferred weights are subject-specific and result in model fitness that exceeds chance. The strongest weights are positive and concentrated within putative brain systems. Collectively, these results set the stage for further explorations of the asymmetric, weighted, and signed network and the implications of the newly defined edge weights for network analyses.

## Modular organization of the asymmetric, weighted, and signed connectome

One of the hallmarks of biological neural networks is that they are organized into densely connected sub-networks called "modules" or "communities"[12]. Although there is a shared correspondence between anatomical modules–defined from streamline-derived structural connectivity–and functional modules, the alignment is inexact[35,36]. Here, we examine the modular structure of anatomical connectivity with the newly derived asymmetric, weighted, and signed connectome and compare its organization with the modular structure derived from a connectome in which edges are weighted based on streamline density.

To detect communities we optimized a signed variant of the modularity quality function[37] using the Louvain algorithm[38]. The output of the algorithm is sensitive to initial conditions and was optimized 1000 times for each of the two weighting schemes. In both cases we fixed the structural resolution parameter to $\gamma = 1$. We aggregated and compared these results by computing coassignment matrices for each connectome, tallying the frequency with which node pairs were assigned to the same module across all 1000 repetitions (Fig. 2a, b). For the sake of visualization, we also calculated consensus communities for each matrix (Fig. 2c, d). We then calculated the difference between the two co-assignment matrices (Fig. 2e). We found that

communities in the asymmetric, weighted, and signed matrix exhibited reduced laterality[39] and tended to span the cerebral hemispheres whereas communities detected using the fiber density matrix tended to be more lateralized (t-test $p < 10^{-15}$; Fig. 2g). We note that these observations were anticipated, given that fMRI BOLD activity was involved in the estimation of structural connection weights and FC exhibits strong homotopic connectivity between left and right cerebral hemispheres.

Next, we asked whether the community structure of the asymmetric, weighted, and signed matrix was better aligned with functional connectivity (correlation structure of resting fMRI BOLD data) than the fiber density matrix and its system-level architecture. To address this question, we imposed canonical brain systems (coarse- and fine-scale intrinsic connectivity networks defined in[34]; Fig. 2f) on each matrix and calculated the induced modularity ($Q^*$). We found that the asymmetric matrix exhibited greater modularity than the fiber density matrix (t-test, $p < 10^{-15}$; Fig. 2h). We also calculated the adjusted Rand index (ARI) between detected partitions and fine- and coarse-scale systems. ARI is a measure of partition similarity; larger values indicate that two partitions are more similar. For both the fine and coarse system partitions, we found that the ARI was greater when compared to partitions detected using the asymmetric and signed matrices than partitions detected using the fiber density matrices (two-sample t-tests; maximum $p < 10^{-15}$; Fig. 2i).

In addition, we conducted a number of supplemental analyses to explore the modular structure of these networks in more detail, providing evidence that they exhibit hierarchical community structure (Fig. S9), and that the modules from our model were more strongly enriched for "brain map"[40] annotations describing properties ranging from density of receptors to the relative expansion of brain areas across development and evolution (Fig. S12). In addition, we introduce a new "geographic" null model for use with modularity maximization (Figs. S11 and S10).

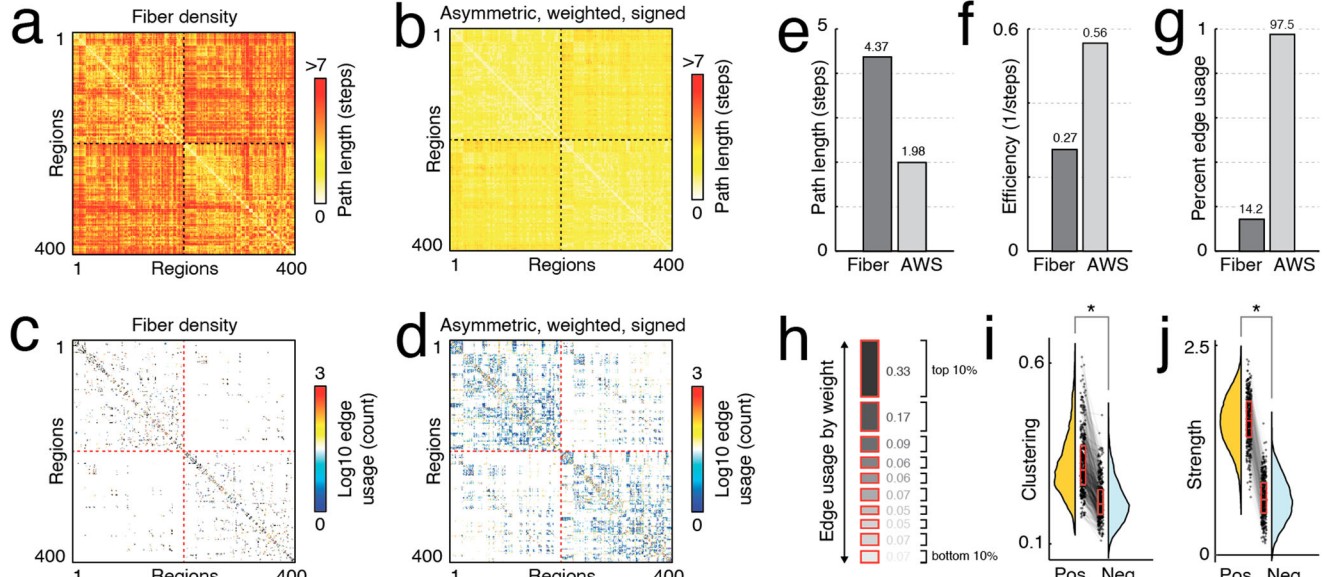

**Fig. 3 | Network statistics for signed, weighted, and asymmetric matrix. a** Path length (number of steps) between all pairs of nodes derived from the fiber density matrix ($C_{ij} = \frac{1}{W_{ij}}$). **b** Path length for new matrix ($C_{ij} = \frac{1}{W_{ij}+1+\varepsilon}$). **c** Edge usage matrix for fiber density network. **d** Edge usage matrix for new matrix. Panels **e**–**g** characteristic path length, and network efficiency, and percentage of edges used in shortest paths between matrices. Panel **h** depicts the breakdown of edge usage by decile–e.g. top decile accounts for 33% of edges used on shortest paths. Panels **i** and **j** compare local clustering and strength (weighted degree) between the

two matrices. We grouped edges into percentiles (deciles; lowest deciles include negative weights) based on their weights and calculated how frequently edges in each decile are involved in shortest paths. Box plots, shown in red and overlaid on data points in **e** and **f**, depict the interquartile range (IQR) and the median value of the distribution. Whiskers extend to the nearest points ± 1.5 × IQR above and below the 25th and 75th percentiles. Asterisks indicate a significant difference between box plots ($n = 400$ for each box plot).

Finally, we also repeated several of the analyses from this and the previous section using mouse anatomical connectivity data made available by the Allen Brain Institute[1] and fMRI data acquired from a cohort of $N = 18$ anaesthetized mice (see Fig. S13 and "Materials and methods" section for more details).

## Graph theoretic properties of the asymmetric, weighted, and signed connectome

In the previous section, we explored the modular architecture of the newly derived asymmetric, weighted, and signed matrix, comparing it with analogous measures made on the fiber density matrix. Modular structure, however, is but one example of a network metric – it assesses a network's organization at the "meso-scale". However, other measures can be meaningfully applied to probe global (whole-network) and local (regional) properties. In this section, we investigate a subset of those measures.

First, we compared shortest-paths structure. Shortest paths in weighted networks refer to the least-costly route from a source node, *s*, to a target node, *t*. Typically, the length or cost of a shortest path is interpreted as a measure of communication capacity[23]; networks where the average shortest path is low (or the average reciprocal shortest path is large) are considered better-suited for communication.

To detect shortest paths we first mapped weights to costs. In the fiber density matrix, this mapping was accomplished by taking the reciprocal of an edge's weight ($cost = \frac{1}{weight}$) before applying a shortest paths algorithm. In the directed and signed network, however, we performed an additional step to rectify edge weights as the shortest paths algorithm is not compliant with negative edges. Briefly, we subtracted $min(\beta_{ij})$ and added $\varepsilon$ to every edge, where $min(\beta_{ij}) = -0.43$ is the smallest (most negative) weight among all edges weights and $\varepsilon = 0.0027$ was the weight of the weakest edge in the fiber density network. This transformation ensures that all existing white-matter edges have weights that are nonzero and positive. Following this transformation, we used the reciprocal transform to map weights to cost.

The shortest paths matrices for both networks are shown in Fig. 3a, b. Strikingly, the number of steps in the least-costly paths was much greater for the fiber density matrix than for the asymmetric, weighted, and signed network (Fig. 3e, f). This likely is a consequence of the heavy-tailed fiber density distribution; because a small number of connections exhibit orders of magnitude stronger weights than the others, the cost of including those edges in shortest path is exceptionally small. From the perspective of the shortest paths algorithm, it is optimal to direct paths through these ultra low-cost edges, possibly even at the expense of direct connections[13,41]. Further evidence for this claim comes from the shortest paths usage; in the fiber density matrix, the fraction of edges that are used in at least one shortest path is only 14.2% (Fig. 3c,d), whereas in the asymmetric, weighted, and signed networks, 97.5% of all edges get used at least once (Fig. 3).

The signed nature of the network means that we can also examine and compare properties of positive and negative edges to one another. That is, we can construct two versions of the same network: one in which nodes are linked via positive connections only and another with negative connections. Interestingly, we find that the positive network exhibits greater local clustering (paired sample *t*-test, $p < 10^{-15}$; Fig. 3i). That is, positive connections tend to form dense triangles and cliques around nodes at a greater rate than negative connections. Additionally, we find that nodes' positive weighted degrees (total weight of all incident positive connections) exceeds that of their negative strength (Fig. 3j).

In summary, we calculate a series of network statistics and show that their values differ, sometimes dramatically, depending on whether we weight edges using our regression-based framework or using more traditional diffusion/imaging parameters. In some specific cases, we find that statistics calculated on the asymmetric, weighted, and signed network are better aligned with our intuition about network function than statistics calculated based on fiber density. Collectively, these results underscore the impact of user decisions on network properties and our interpretation of network organization and function.

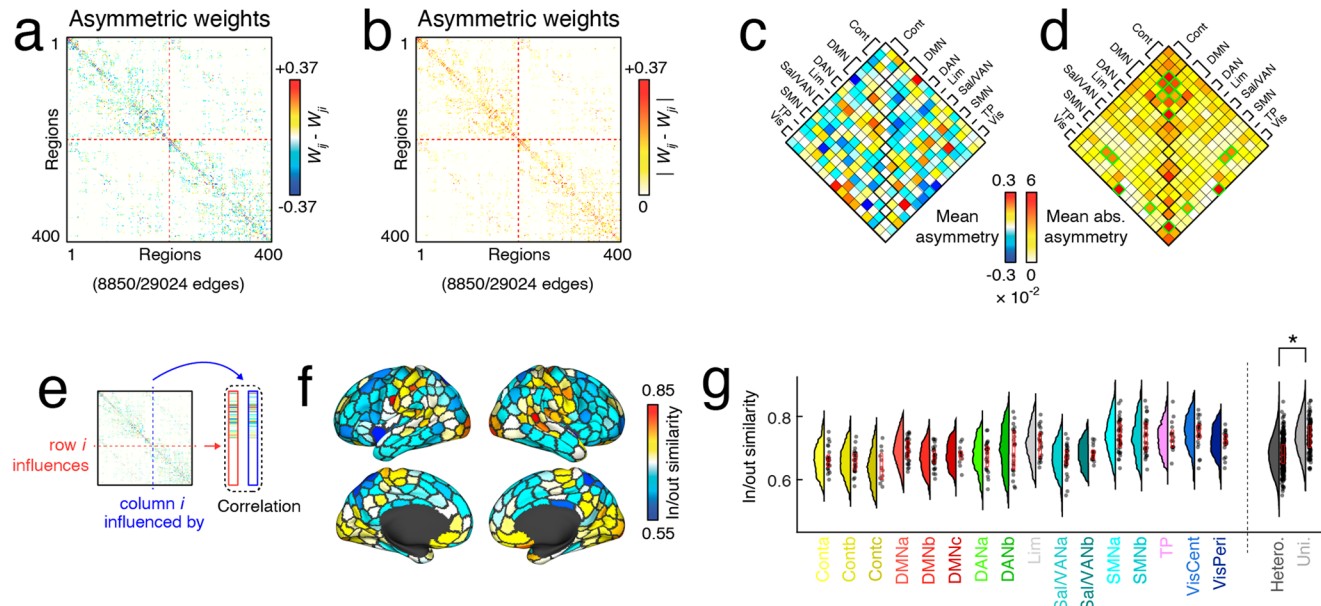

**Fig. 4 | Asymmetries of influence between brain regions as assessed by inferred structural weights. a** Significant asymmetries, and (**b**) absolute value of significant asymmetries were reorganized into system by system matrices (**c**) and (**d**) respectively. Note the increased absolute asymmetries within functionally defined systems in panel **d**. As illustrated in the schematic in panel **e** next we measured in/out similarity as the correlation between incoming and outgoing weights per region. **f** Here we show an example of in/out similarity plotted to the brains surface. **g** Finally, we plot the per-system distribution of in/out similarity values across subjects. Boxplots on the right divide these systems into unimodal and hetero-modal regions to show that there is more in/out similarity in unimodal systems. Box plots, shown in red and overlaid on data points in e and f, depict the interquartile range (IQR) and the median value of the distribution. Whiskers extend to the nearest points ± 1.5 × IQR above and below the 25th and 75th percentiles. Asterisks indicate a significant difference between box plots (Box plots $N = 24, 25, 12, 34, 32, 13, 27, 25, 24, 34, 17, 39, 31, 16, 24, 23, 283, 117$).

## Asymmetries in connection weights

Due to technological limitations, structural connection weights estimated in vivo using diffusion imaging and tractography methods lack directionality–i.e. $W_{ij} = W_{ji}$. Here, however, the regression framework we use allows for asymmetries, such that the weights of incoming and outgoing connections can deviate from one another. In this section, we describe a select set of asymmetries in greater detail.

We measured asymmetry using a simple statistical test. Specifically, we identified pairs of regions whose weights were consistently asymmetric across 95 Human Connectome Project participants. This involved fitting weights independently for each subject and edge, and for every pair of nodes $i$ and $j$, identifying connections where the distribution of asymmetry values, $W_{ij} - W_{ji}$, excluded zero (false discovery rate fixed at $q = 0.05$ resulting in $p_{adj} = 0.015$; Fig. 4a, b)[42]. We found that, of 29024 possible connections, 8850 (approximately 30%) exhibited significant asymmetries in terms of their incoming vs outgoing weights.

Next, we asked whether edges whose weights were significantly asymmetric were preferentially concentrated within or between specific brain systems (Fig. 4c, d). To assess whether this was the case, we created a "mask" of edges that exhibited statistically significant asymmetries and aggregated (summed) these connections within and between every pair of systems. We performed this procedure first using the signed difference in edge weights and again using the absolute difference. These summed values were compared against a null distribution generated via a geometry-preserving null model[43]. We found that a number of system pairs exhibited greater than expected asymmetries, including connections that fall within control, default, and visual networks (ContC, DMNa, DMNc, central visual), as well as connections that fall between systems (ContC-DMNa, DMNa-DMNb, temporo-parietal and both ContC and DMNa, central visual and SMNb, and peripheral visual with DANb).

As a second measure of asymmetry, we compared the weights of nodes' incoming and outgoing connection profiles – the extent to which its activity is predicted by vs predicts the activity of its neighbors. To do this, we calculated the linear product-moment correlation between vectors associated with row and column $i$ in the asymmetric, weighted, and signed connectivity matrix (Fig. 4e). This procedure resulted in a single similarity score (correlation) for each brain region. In general, we found that in-out similarity was region-specific and varied between putative brain systems (Fig. 4f), with regions in sensorimotor systems exhibiting greater in/out similarity (Fig. 4g). Indeed, when we grouped systems based on unimodal (visual + somatomotor) and heteromodal (all other systems) labels, we found that unimodal systems exhibited greater similarity (two-sample $t$-test, $p < 10^{-15}$; Fig. 4h).

In an additional analysis, we also identified node pairs, $i$ and $j$, where sign($W_{ij}$) ≠ sign($W_{ji}$) (Fig. S14). We performed this analysis at the level of individual subjects and calculated the proportion of edges with an asymmetry of sign that fall either within or between brain systems (Fig. S14a). We repeated this analysis for every individual and found that between-system edges were more likely to exhibit an asymmetry of sign than within-system edges (two-sample $t$-test, $p < 10^{-15}$; Fig. S14b).

Finally, we performed an additional analysis to probe the importance of the asymmetries in our model for model performance. We found that forcing the weights of our trained model to be symmetric degraded model performance for both group-based assessments in mice (Fig. S17b) and humans (Fig. S17c) as well as individual based models (Fig. S17d, g–i). Additionally, we found that the regions whose predictions were most effected by forcing symmetry in connection weights were heteromodal regions(Fig. S17e).

Collectively, these results suggest that local asymmetries are well circumscribed by canonically defined brain systems. Additionally, our results suggest that asymmetry in regional incoming and outgoing connection weights run along a unimodal-heteromodal axis.

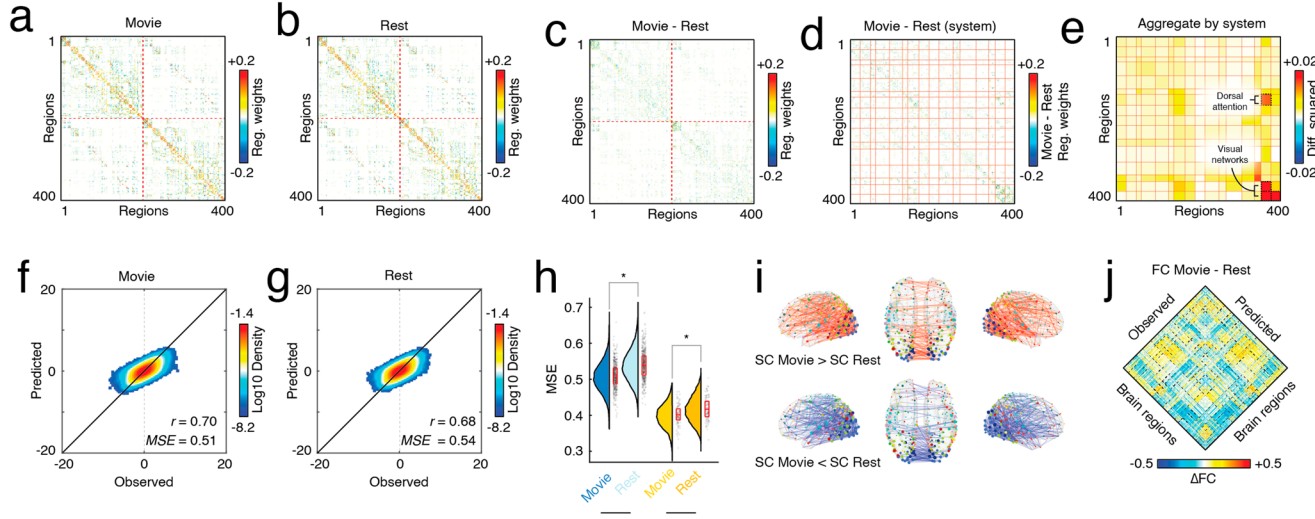

**Fig. 5 | Comparing matrices fit to resting-state and movie-watching data.** We analyzed 7T data from 117 participants in the Human Connectome Project. Using the same binary mask as in the previous sections, we fit edge weights for both conditions at the group level (pooling time series data across all subjects/scans) and individual level (pooling data from the same subjects). Weights for (**a**) group-level movie-watching state matrix and (**b**) group-level resting-state matrix. Panels **c** and **d** show the difference in edge weights (movie minus rest); rows and columns in panel **c** are ordered identically to panels **a** and **b**, whereas in panel **d** rows and columns are reordered by brain systems. **e** The average weight across existing connections between every pair of systems. Here, weights were squared prior to averaging. Panels **f** and **g** depict two-dimensional histograms of observed and predicted activity. **h** Model fitness when fit to pooled, group-level data (left) and individual data (right). Box plots, shown in red and overlaid on data points in **e** and **f**, depict the interquartile range (IQR) and the median value of the distribution. Whiskers extend to the nearest points ± 1.5 × IQR above and below the 25th and 75th percentiles. Asterisks indicate a significant difference between box plots (Group: $N = 468$, Individual: $N = 117$). **i** Edges whose difference between movie and rest are significantly greater or less than zero plotted in anatomical space. **j** Differences in functional connectivity (FC) between movie and rest for both the observed data (left) and the predicted (right).

## Weights are modulated by state

In most connectome studies, weights are defined based on streamline or tractographic properties. These measures are typically assumed to be invariant over short timescales. However, the asymmetric, weighted, and signed connectome uses functional imaging data, which does fluctuate in response to stimulation, to assign weights to structural connections. A natural question to ask is whether these weights are state-dependent and, if so, whether we can detect changes in connections' weights as a function of in-scanner task and whether those changes can be localized to specific tracts or brain systems.

To address these questions, we used resting-state and movie-watching data from the Human Connectome Project's 7T dataset, focusing on a subset of 117 participants whose data passed quality checks and for whom all four scans were available. Using the same SC binary mask, we fit edge weights at the group level, pooling data across all subjects and scans to generate two asymmetric, weighted, and signed matrices: one based on resting-state and the other based on movie-watching data (Fig. 5a, b). In parallel, we also fit models at the level of individual subjects, pooling scans from the same subject to generate estimates of resting-state and movie-watching edges weights. In both cases, we found comparable performance (mean squared error) between both rest and movie data, with movies exhibiting slightly better performance than resting state (paired sample $t$-test; subject-level, $p < 10^{-15}$; group-level $p = 1.2 × 10^{-12}$; Fig. 5f–h).

First, we calculated the difference between edge weights for each subject and averaged the differences across subjects (Fig. 5c, d). At each edge, we performed a paired-sample $t$-test on the differenced edge weight distributions. We found that, out of $m = 29,204$ total edges, 2463 exhibited significant state-dependent differences (multiple comparisons controlled for by fixing false discovery rate at $q = 0.01$ and adjusting the critical $p$-value, $p_{adj} = 8.5 × 10^{-4}$). Although these edges were distributed across the entire brain, they were significantly concentrated within a small subset of systems (Fig. 5e; dashed black borders around system blocks). Specifically, we found significant

system-level effects within central and peripheral visual networks, from edges in the central visual network to the peripheral visual network (but not vice versa, and from the dorsal attention network (DANa) to the central visual network (spin test, false discovery rate fixed at $q = 0.01$, $p_{adj} = 1.6 × 10^{-4}$).

Projected into anatomical space, we find that, as expected, the connections that differ from rest to movie-watching tend to involve regions in visual networks (Fig. 5i). Interestingly, there are approximately as many connections whose weights increase from rest to movies as there are those that decrease, an effect that holds both within the visual networks (241 increases vs 218 decreases) but also across the entire brain (1292 increases vs 1171 decreases).

## Differences in the weighted, signed, and directed connectome across the human lifespan

To this point, we have estimated the weights of asymmetric, weighted, and signed structural connections, described properties of the resulting network, exposed asymmetries in connections' weights, and demonstrated that the weights are systematically modulated by task (rest vs movie). In this section, we investigate individual differences in connections' weights and associate them with differences across the human lifespan (7-85 years).

To do so, we used data from the Nathan Kline Institute's enhanced Rockland Sample[44], which included both diffusion weighted and functional MRI data for $N = 542$ participants. In-scanner head movement is known to vary systematically with age. To address motion-related concerns, we adopted the same conservative procedure as reported in[45] for motion censoring. Specifically, for each of the remaining subjects, we dropped frames in which motion exceeded a pre-defined threshold (FD > 0.15 mm). We also dropped time points that were within two frames of any supra-threshold frame or failed for a contiguous sequence of five frames or more. Following this procedure, we excluded any participant for whom the fraction of retained frames was fewer than 50% of their total number of frames.

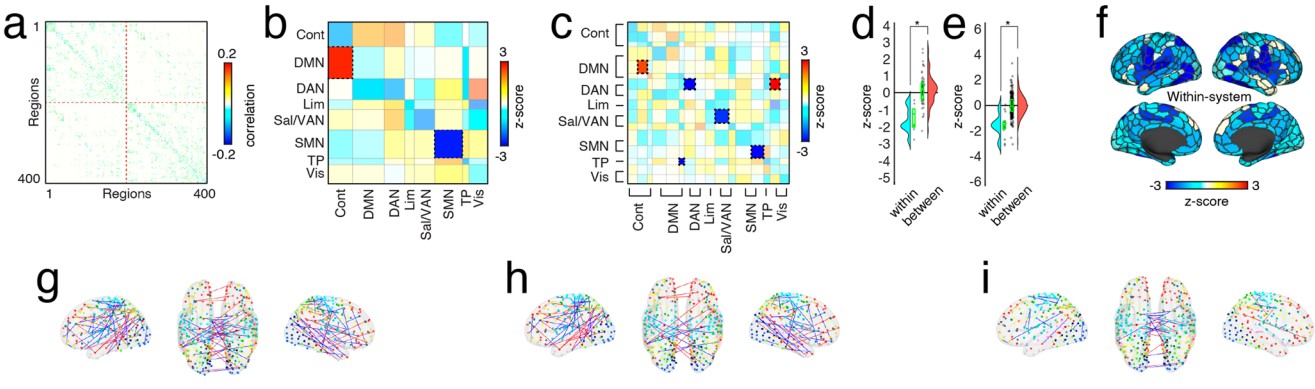

**Fig. 6 | Age-related differences in inferred structural weights are more common within functional brain systems. a** Edge level correlations with age. **b, c** Matrices of significant results for the coarse- and fine-scale system spin tests, respectively (Cont = Control, DMN = default mode network, DAN = dorsal attention network, Lim = limbic, Sal/VAN = salience, SMN = somatomotor, TP = temporoparietal, Vis = visual). **d, e** Boxplots comparing the z-scores from the spin-test null model with the real values. One boxplot displays within system z-score values, and the other displays between system z-score values. Box plots, shown in red and overlaid on data points in **e** and **f**, depict the interquartile range (IQR) and the median value of the distribution. Whiskers extend to the nearest points ±1.5 × IQR above and below the 25th and 75th percentiles. Asterisks indicate a significant difference between box plots (two-sample t-test; panel **d**: within $N = 7$, between $N = 42$; panel e: within $N = 16$, between $N = 240$). **f** Within-system z-scores from the fine-scale system spin test mapped onto cortical parcels. Panels **g**–**i** show individual edges that exhibit age-related correlations ($p < 0.05$; uncorrected). Blue and red edges correspond to edges whose weights decrease or increase with age, respectively. Panels **g**–**i** are presented for the sake of visualization only.

Collectively, these procedures left $N = 474$ participants with high-quality (low-motion) data for further analysis.

For each subject, we used the regression-based framework to fit weights to every structural connection, generating subject-specific asymmetric, weighted, and signed matrices. Note that, here, we restricted every subject to have the same binary set of consensus edges estimated from subjects aged 18-35 years; only edges' weights varied across individuals. Prior to calculating age-related differences, we regressed out of each edge the following variables: sex (binary variable), gray matter volume, mean framewise displacement of all "low-motion" frames, and number of frames dropped due to motion contamination. Finally, using the residuals from this procedure, we calculated their linear product-moment correlation with subjects' biological ages resulting in a (sparse) matrix of age-related correlations (Fig. 6a). For the sake of visualization, we show subsets of edges that pass uncorrected statistical tests (Fig. 6g–i).

Previous studies have found that age-related differences in functional connectivity respect putative system boundaries[46–48]. Accordingly, we performed statistical tests at the level of systems[34]. Specifically, we calculated the mean correlation of all edges that fell between/within every pair of systems. We then compared these observed values with null distributions generated using "spin tests" (1000 repetitions). System pairs for whom the observed correlation exceeded that of the null were considered statistically significant (false discovery rate fixed at $q = 0.05$; $p_{adj} = 0.0088$). In line with previous work, we found that age-related decreases in connection weight tended to concentrate within brain systems, whereas between-system weights were, generally, centered around a value of zero (two-sample t-test, $p = 7.65 × 10^{-13}$; Fig. 6b–f). More specifically, we found that connections within the somatomotor network significantly decreased their weight with age while connections from the default mode to the control network increased (Fig. 6b; this was at the coarse scale). Repeating this analysis at a finer scale allowed us to better localize those effects. In particular, we found significant increases in connection weight from default mode B to control B, as well as an increase in connection weights from dorsal attention network A to the central visual module – an effect that had been previously obscured at the coarse scale. We also detected significant decreases in connection weight with age, concentrated within somatomotor network B, as well as previously undetected decreases within dorsal attention network A,

salience/ventral attention network A, and from the temporo-parietal network to default mode C.

Altogether, these findings recapitulate well-known age effects that had been previously reported using functional connectivity data. However, our approach grounds these effects in anatomical connectivity, forming a multi-modal bridge between studies of anatomical and functional age-related differences and opening up avenues for future applied studies.

## Discussion

The correct weighting of structural connections is not known. Most strategies for assigning edge weights do so based on microstructural or tractographic parameters. Though commonly used, the values of these parameters are, in general, misaligned with the interpretation that weights reflect the usage of inter-regional anatomical connections. This and other limitations–e.g. inability to detect directed connectivity–motivate the exploration of alternative weighting schemes, including multi-modal syntheses of structural, diffusion, and functional imaging features.

We note that our approach shares some features with existing frameworks–e.g. effective connectivity and dynamic causal models[49–54]–in that it returns directed connections. However, our weighting scheme is not generative–i.e. it cannot be used to generate new synthetic time series data. Additionally, our approach is not seeking to solve the inverse problem–i.e. inferring structure from function. Rather, it is explanatory and represents a means of weighting already reconstructed fiber tracts. It is therefore distinct from extant approaches in network neuroscience, and presents opportunities for multiple follow-up studies and applications in other neuroscientific disciplines.

Over the past two decades, network neuroscience has led to a number of discoveries about the organization of brain networks. These include small-worlds[9], hubs[10] and rich clubs[11], modules and communities[12], and cost-efficient wiring[55]. These canonical findings have been observed not only in human brain networks reconstructed at the macroscale, but have been reported across phylogeny and at all spatial scales[56].

Despite this preponderance of converging evidence, the specifics of these findings often depend critically on whether or not to weight edges and the precise measure used. Consider the prototypical "small

world" model of the brain, where long-distance edges are thought to represent shortcuts that allow signals to propagate long distances in relatively few hops. Although both weighted and binary networks exhibit small-world properties–relatively short path length and strong local clustering–when edges are weighted based on streamline density, the strong distance-dependence of streamlines ensures that virtually no long-distance connections are included in the network's shortest path structure[13].

The example cited above represents just one instance where a processing decision (whether and how to weight structural connections) leads to different interpretations of brain network function, *vis a vis* role of long-distance connections in interareal communication. Interestingly, we make a similar observation here; when edges are weighted with regression coefficients and their signs rectified, we recover a shortest path structure in which most edges contribute to at least one shortest path and modules that are now better aligned with functional brain systems[57]. Our work even presents challenges for how we define connectomes[58]. When we refer to the connectome, we often imagine an enumerable and finite set of neural elements and connections[59,60]. Here, however, edge weights are context dependent, impermanent, and vary with task/cognitive state. Essentially, the structural edges inherit features usually reserved for functional connections, placing our approach slightly at odds with the perspective that structural connections are fixed over short timescales (duration of typical scan session).

More importantly, the secrets of the connectome are far from unlocked. Existing data and methods have not unambiguously mapped structure to function, for example, whether graph-theoretic measures can unveil functional properties of a brain (and which properties, specifically) is not clearly elucidated, and, although many studies have identified statistical associations between network properties and clinical, cognitive, behavioral, and developmental markers, the mechanisms that underlie those associations are largely unknown. Collectively, this motivates further neuroscientific exploration, both of new data/connectivity modalities[61] and experimental paradigms, as well as methodological frameworks.

The edge-weighting scheme that we explore here can be used to help understand one of the central questions of network neuroscience: how does the brain's anatomical connectivity constrain its function? Past studies have established a link between structural and functional connectivity[35]. Empirical findings have shown that insults to structural connections cause acute loss or reorganization of functional connectivity[62]. Even in intact brains, structural connection weights are correlated with their functional analogs and pairs of brain regions that are connected directly or via few processing steps have proportionally stronger FC[10]. In parallel, in silico dynamical systems models have used anatomical connectivity to constrain simulated brain activity[63,64], generating synthetic fMRI data whose correlation structure can be compared with empirical FC or analytic estimates of interregional communication capacity[23,65].

Here, rather than compare SC to FC, we incorporate functional information directly into the estimates of edge weights[66–69]. This process generates a singular network object whose fitness (a metric that, itself, can be interpreted as a measure of structure-function coupling) can be estimated globally as the total error between observed and predicted activity[20] or parsed into local (regional) error terms, analogous to recent approaches for linking anatomical and functional connectivity weights[45,70]. We note, however, that this approach is also distinct from most studies that report structure-function correspondence, in that we seek a set of parameters that maximizes that correspondence, whereas most studies report a correlation between functional data (FC or activity) and structural networks and their derivatives[71].

Throughout the study, we compare properties of the asymmetric, weighted, and signed network with a network in which edge weights

represent fiber densities – a more traditional measure of structural connectivity. In general (and unsurprisingly), the properties of these networks are often dissimilar. That is, how we choose to weight a network's edges can change its graph-theoretic profile and impact how we might interpret its function. Although we remain agnostic as to which weighting scheme is superior, we note that the asymmetric network both outperforms the fiber density network on a number of applications, and has properties that are better aligned with intuition[13].

For instance, we find that the modular structure of the asymmetric network tends to be less lateralized–i.e. modules are more likely to contain nodes from both hemispheres–than the fiber density network. This observation suggests that the reweighting of the network helps circumvent one of the peculiarities (or limitations) of community-detection methods applied to structural brain networks. Namely, because fiber densities and weights derived from tract-tracing experiments tend to be heavy-tailed and distance-dependent[13,28] and because long-distance interhemispheric tracts are notoriously challenging to reconstruct from diffusion imaging data[72,73], communities tend to be spatially contiguous and exhibit poor correspondence with systems/communities derived from functional recordings[35].

Additionally, we take advantage of recent advances in neuroinformatics to compare communities with brain maps–i.e. the regional or vertex-wide expression of genetic, transcriptomic, evolutionary, and developmental markers. We find that communities obtained from the asymmetric network tend to be significantly enriched for many of these markers to an extent above and beyond the communities obtained from the fiber density matrix. These observations suggest that the multi-modal network generated by endowing structural connections with functionally relevant information tightens the link between network organization and brain-based markers.

Two of the unique features of the networks we construct here are that edge weights are signed and directed. How do we interpret these features? Are there possible neurophysiogical explanations?

At their core, the edge weights estimated here are statistical constructs. Specifically, they represent how well the past activity of node $i$'s connected neighbors explains $i$'s future activity. For a given node $i$, its in-weights must also be interpreted as a group; they are estimated simultaneously, with neighbors of $i$ competing for the pool of unexplained variance. In a simple two node system, a negative weight from node $i$ to $j$ indicates that when the activity of $i$ increases, the activity of $j$ tends to decrease proportionally at the next time point. As you increase the number of connected nodes, negative weights should be interpreted in the context of the neighbors of the node whose activity they are predicting. Similarly, asymmetric weights arise from modeling future activity from the perspective of different nodes, such that node $i$ has greater influence over node $j$'s model than the reverse. This interpretation in which the connection, $W_{j \to i}$, represents how much the history of $j$ explains the current state of $i$ is broadly in line with extant modeling frameworks for estimating effective connectivity[74].

Although it is tempting to ascribe "excitatory" and "inhibitory" labels to positive and negative edge weights, this terminology is typically reserved for cell-to-cell projections. In general, the neurochemical (e.g. glutamate/excitatory and GABA/inhibitory) contributions to the diffusion MRI and fMRI BOLD signal are not easily parsed[75]. However, we can speculate about possible underlying mechanisms that support signed edges in large-scale networks. One possible explanation involves feed-forward or feedback inhibition[76], whereby excitatory inter-regional connections cause inhibition in their target region either by directly exciting local inhibitory interneurons, or by exciting local interneurons indirectly through connecting pyramidal cells. Indeed, recent studies have suggested that the balance of glutamate/GABA underlies the antagonistic (anti-correlated) activity of large-scale brain systems[77].

Similarly, although it is tempting to ascribe a causal story to the asymmetric weights of our model such that one region has more causal

influence over another region, causality is notoriously difficult to assess, especially in complex systems like the brain[78,79]. Many methods have been developed in attempt to disambiguate causal asymmetries in brains including direct electrical stimulation of brain regions[80] and effective connectivity and dynamic causal models[49–54]. Still, the difficulty involved in characterizing causality in complex systems warrants a multi-pronged approach. With the asymmetric, weighted and signed connectome, we hope to provide an additional, easily calculated metric by which to assess asymmetric relationships between brain regions. With multiple metrics we might be able to better triangulate on the reality of causal asymmetries in these systems.

Indeed, in a supplementary analysis we found that the asymmetric weights estimated using our model are uncorrelated with weights estimated from tract-tracing data (which offers an anatomical perspective on asymmetric connectivity)[1]. These observations suggest that incorporating functional information may yield largely complementary information about brain connectivity to traditional edge weight metrics. In support of this, a number of our findings suggest that the weights in our model are not randomly distributed throughout the brain, but reflect known organizational axes of brain networks. For instance, asymmetric edges with different signs (+/−) tend to fall between large-scale brain systems rather than within and the similarity of in-strength and out-strength is lowest in heteromodal poles, suggesting that asymmetry might be a hallmark of polyfunctionality.

Irrespective of their underlying origins, the signed and asymmetric edges in the networks constructed here exhibit non-random organization in terms of their distribution across canonical brain systems and relationship to other network/geometric measures–e.g. clustering coefficient and fiber length. These features, of course, have implications for traditional network analyses and may spur methodological innovation within the field of network neuroscience. For instance, interregional communication models rely on shortest paths and diffusion dynamics to estimate communication efficacy[23,81]. However, these measures are not well-defined for networks with signed edges. Here, we circumvent this issue by offsetting edge weights, forcing negative connections to have small (but positive) values. However, there are likely many alternative strategies that embrace the signed nature of edges that could be explored in future studies[82].

This study has a number of limitations. Most notably, the primary results rely on the use of diffusion-weighted MRI and tractography for reconstructing white-matter tracts. Although these methods are still used widely, they have well-documented drawbacks and biases that call into questions the verisimilitude of structural connectivity networks[83,84]. We partially mitigate these concerns by replicating our findings using tract-tracing data made available through the Allen Brain Institute[1]. Unlike tractography, in which anatomical connections are inferred non-invasively, antero-/retrograde tract-tracing is considered the gold standard mapping large-scale connectivity[85]. We expect that advances in imaging, tractography algorithms[86,87], and better alignment of multi-scale datasets will narrow the gap between tract-tracing and tractography in future studies[88].

Another possible limitation of the current study is its focus on neocortex only. Doing so necessarily ignores contributions from subcortical and cerebellar regions when modeling node-level time series. Adding additional regions is not computationally prohibitive and in principle could be addressed easily.

Our study also presents a number of opportunities for future studies. Among the most obvious is the empirical validation of edge weights estimated here. Datasets in which stimulation is paired with brain-wide recordings make it feasible to estimate directed influence between brain stimulus-target pairs of regions[89,90]. These estimates could be compared directly to the connection weights inferred here. Relatedly, our results should be compared against those obtained using other methods for inferring asymmetric and effective connectivity[42,91].

Our preliminary findings using movie-watching data suggest that our weighting scheme may be suitable for detecting state-specific changes in structural connection weights. Future studies should explore the sensitivity of this approach for other state-based comparisons. We note that, unlike unthresholded functional connectivity, which, when used in a brain-wide association or case-control study results in $N(N-1)/2$ comparisons, our approach results in much fewer connections[92]. Statistical tests only need to be performed for existing structural connections, possibly increasing the statistical power of these types of studies[93].

## Methods

In this section, we describe all four datasets that we analyzed. Briefly, they include three human MRI datasets: two from the Human Connectome Project and another from the Nathan Kline Institute. In addition, we also analyzed tract-tracing and functional MRI data from mice.

### Datasets: Human Connectome Project 3T resting-state and diffusion-weighted MRI

The Human Connectome Project (HCP) 3T dataset[29] consists of structural magnetic resonance imaging (T1w), functional magnetic resonance imaging (fMRI), and diffusion magnetic resonance imaging (dMRI) young adult subjects, some of which are twins. Here we use a subset of the available subjects. These subjects were selected as they comprise the "100 Unrelated Subjects" released by the Connectome Coordination Facility. After excluding data based on completeness and quality control (4 exclusions based on excessive framewise displacement during scanning; 1 exclusion due to software failure), the final subset included 95 subjects (56% female, mean age = 29.29 ± 3.66, age range = 22–36). The study was approved by the Washington University Institutional Review Board and informed consent was obtained from all subjects.

A comprehensive description of the imaging parameters and image prepocessing can be found in ref. 94. Images were collected on a 3T Siemens Connectome Skyra with a 32-channel head coil. Subjects underwent two T1-weighted structural scans, which were averaged for each subject (TR = 2400 ms, TE = 2.14 ms, flip angle = 8°, 0.7 mm isotropic voxel resolution). Subjects underwent four resting-state fMRI scans over a two-day span. The fMRI data was acquired with a gradient-echo planar imaging sequence (TR = 720 ms, TE = 33.1 ms, flip angle = 52°, 2 mm isotropic voxel resolution, multiband factor = 8). Each resting-state run duration was 14:33 min, with eyes open and instructions to fixate on a cross. Subjects underwent 14 task fMRI scans over a two-day span. The fMRI data was collected with the same sequence parameters as the resting-state fMRI. The fMRI runs consisted of working memory (5:01 min, 405 frames), gambling (3:12, 253), motor (3:34, 284), language (3:57, 316), social cognition (3:27, 274), relational processing (2:56, 232), and emotional processing (2:16, 176) tasks. Finally, subjects underwent two diffusion MRI scans, which were acquired with a spin-echo planar imaging sequence (TR = 5520 ms, TE = 89.5 ms, flip angle = 78°, 1.25 mm isotropic voxel resolution, b-vales = 1000, 2000, 3000 s/mm², 90 diffusion weighed volumes for each shell, 18 b = 0 volumes). These two scans were taken with opposite phase encoding directions and averaged.

Structural, functional, and diffusion images were minimally preprocessed according to the description provided in[94], as implemented and shared by the Connectome Coordination Facility. Briefly, T1w images were aligned to MNI space before undergoing FreeSurfer's (version 5.3) cortical reconstruction workflow, as part of the HCP Pipeline's PreFreeSurfer, FreeSurfer, and PostFreeSurfer steps. Functional images were corrected for gradient distortion, susceptibility distortion, and motion, and then aligned to the corresponding T1w with one spline interpolation step. This volume was further corrected for intensity bias and normalized to a mean of 10000. This volume was then projected to the 2mm *32k_fs_LR* mesh, excluding outliers, and

aligned to a common space using a multi-modal surface registration[95]. The resultant CIFTI file for each HCP subject used in this study followed the file naming pattern: `*_Atlas_-MSMAll_hp2000_clean.dtseries.nii`. These steps are performed as part of the HCP Pipeline's fMRIVolume and fMRISurface steps. Each minimally preprocessed fMRI was linearly detrended, band-pass filtered (0.008-0.008 Hz), confound regressed and standardized using Nilearn's `signal.clean` function, which removes confounds orthogonally to the temporal filters. The confound regression strategy included six motion estimates, mean signal from a white matter, cerebrospinal fluid, and whole brain mask, derivatives of these previous nine regressors, and squares of these 18 terms. Spike regressors were not applied. Following these preprocessing operations, the mean signal was taken at each time frame for each node, as defined by the Schaefer 200 parcellation[34] in *32k fs_LR* space. Diffusion images were normalized to the mean b0 image, corrected for EPI, eddy current, and gradient non-linearity distortions, and motion, and aligned to subject anatomical space using a boundary-based registration as part of the HCP pipeline's Diffusion Preprocessing step. In addition to HCP's minimal preprocessing, diffusion images were corrected for intensity non-uniformity with `N4BiasFieldCorrection`[96]. The Dipy toolbox (version 1.1)[97] was used to fit a multi-shell multi-tissue constrained spherical deconvolution[98] to the data with a spherical harmonics order of 8, using tissue maps estimated with FSL's `fast`[99]. Tractography was performed using Dipy's `Local Tracking` module[97]. Multiple instances of probabilistic tractography were run per subject[100], varying the step size and maximum turning angle of the algorithm. Tractography was run at step sizes of 0.25 mm, 0.4 mm, 0.5 mm, 0.6 mm, and 0.75 mm with the maximum turning angle set to 20°. Additionally, tractography was run at maximum turning angles of 10°, 16°, 24°, and 30° with the step size set to 0.5 mm. For each instance of tractography, streamlines were randomly seeded three times within each voxel of a white matter mask, retained if longer than 10 mm and with valid endpoints, following Dipy's implementation of anatomically constrained tractography[101], and errant streamlines were filtered based on the cluster confidence index[102]. For each tractography instance, streamline count between regions-of-interest were normalized by dividing the count between regions by the geometric average volume of the regions. Since tractography was run nine times per subject, edge values were collapsed across runs. To do this, the weighted mean was taken with weights based on the proportion of total streamlines at that edge. This operation biases edge weights towards larger values, which reflect tractography instances better parameterized to estimate the geometry of each connection.

### Datasets: Human Connectome Project 7T resting-state and movie-watching data

The Human Connectome Project (HCP) 7T dataset[29] consists of structural magnetic resonance imaging (T1w), resting-state functional magnetic resonance imaging (rsfMRI) data, movie-watching functional magnetic resonance imaging (mwfMRI) from 184 adult subjects. These subjects are a subset of a larger cohort of approximately 1200 subjects additionally scanned at 3T. Subjects' 7T fMRI data were collected during four scan sessions over the course of two or three days at the Center for Magnetic Resonance Research at the University of Minnesota. Subjects' 3T T1w data collected at Washington University in St. Louis. The study was approved by the Washington University Institutional Review Board and informed consent was obtained from all subjects.

We analyzed MRI data collected from $N_s = 129$ subjects (77 female, 52 male), after excluding subjects with poor quality data. Our exclusion criteria was as follows: where each spike is defined as relative framewise displacement of at least 0.25 mm, we excluded subjects who fulfill at least 1 of the following criteria: greater than 15% of time points spike, average framewise displacement greater than 0.2 mm; contains any

spikes larger than 5mm. Following this filter, subjects who contained all four scans were retained. At the time of their first scan, the average subject age was $29.36 \pm 3.36$ years, with a range from 22 to 36. 70 of these subjects were monozygotic twins, 57 were non-monozygotic twins, and 2 were not twins.

A comprehensive description of the imaging parameters and image preprocessing can be found in[94] and in HCP's online documentation (https://www.humanconnectome.org/study/hcp-young-adult/document/1200-subjects-data-release). T1w were collected on a 3T Siemens Connectome Skyra scanner with a 32-channel head coil. Subjects underwent two T1-weighted structural scans, which were averaged for each subject (TR = 2400 ms, TE = 2.14 ms, flip angle = 8°, 0.7 mm isotropic voxel resolution). fMRI were collected on a 7T Siemens Magnetom scanner with a 32-channel head coil. All 7T fMRI data was acquired with a gradient-echo planar imaging sequence (TR = 1000 ms, TE = 22.2 ms, flip angle = 45°, 1.6 mm isotropic voxel resolution, multi-band factor = 5, image acceleration factor = 2, partial Fourier sample = 7/8, echo spacing = 0.64 ms, bandwidth = 1924 Hz/Px). Four resting-state data runs were collected, each lasting 15 minutes (frames = 900), with eyes open and instructions to fixate on a cross. Four movie-watching data runs were collected, each lasting approximately 15 minutes (frames = 921, 918, 915, 901), with subjects passively viewing visual and audio presentations of movie scenes. Movies consisted of both freely available independent films covered by Creative Commons licensing and Hollywood movies prepared for analysis[103]. For both resting state and movie-watching data, two runs were acquired with posterior-to-anterior phase encoding direction and two runs were acquired with anterior-to-posterior phase encoding direction.

Structural and functional images were minimally preprocessed according to the description provided in[94], as implemented and shared by the Connectome Coordination Facility. Briefly, T1w images were aligned to MNI space before undergoing FreeSurfer's (version 5.3) cortical reconstruction workflow, as part of the HCP Pipeline's Pre-FreeSurfer, FreeSurfer, and PostFreeSurfer steps. 7T fMRI images were downloaded after correction and reprocessing announced by the HCP consortium in April, 2018. fMRI images were corrected for gradient distortion, susceptibility distortion, and motion, and then aligned to the corresponding T1w with one spline interpolation step. This volume was further corrected for intensity bias and normalized to a mean of 10000. This volume was then projected to the 2mm *32k fs_LR* mesh, excluding outliers, and aligned to a common space using a multi-modal surface registration[95]. The resultant CIFTI file for each HCP subject used in this study followed the file naming pattern: `*_Atlas_MSMAll_hp2000_clean.dtseries.nii`. These steps are performed as part of the HCP Pipeline's fMRIVolume and fMRISurface steps. Resting state and moving watching fMRI images were nuisance regressed in the same manner. Each minimally preprocessed fMRI was linearly detrended, band-pass filtered (0.008−0.25 Hz), confound regressed and standardized using Nilearn's `signal.clean` function, which removes confounds orthogonally to the temporal filters. The confound regression strategy included six motion estimates, mean signal from a white matter, cerebrospinal fluid, and whole brain mask, derivatives of these previous nine regressors, and squares of these 18 terms. Spike regressors were not applied. Following these preprocessing operations, the mean signal was taken at each time frame for each node, as defined by the Schaefer 400 parcellation[34] in *32k fs_LR* space.

### Datasets: Nathan Kline Institute, Enhanced Rockland Sample 3T resting-state, and diffusion-weighted MRI

The Nathan Kline Institute Rockland Sample (NKI) dataset consisted of structural magnetic resonance imaging, resting-state functional magnetic resonance imaging data, as well as diffusion magnetic resonance imaging data from 811 subjects (downloaded December 2016 from the INDI S3 Bucket) of a community sample of participants across the

lifespan[44]. After excluding subjects based on data and metadata completeness and quality control, the final subset utilized included 542 subjects (56% female, age range = 7–84). The study was approved by the Nathan Kline Institute Institutional Review Board and Montclair State University Institutional Review Board and informed consent was obtained from all subjects. A comprehensive description of the imaging parameters can be found online at the NKI website.

Briefly, images were collected on a Siemens Magneton Trio with a 12-channel head coil. Subjects underwent one T1-weighted structural scan (TR = 1900 ms, TE = 2.52 ms, flip angle = 9°, 1 mm isotropic voxel resolution). Subjects underwent three differently parameterized resting state scans, but only one acquisition is used in the present study. The fMRI data was acquired with a gradient-echo planar imaging sequence (TR = 645 ms, TE = 30 ms, flip angle = 60°, 3 mm isotropic voxel resolution, multiband factor = 4). This resting state run lasted approximately 9:41 seconds, with eyes open and instructions to fixate on a cross. Subjects underwent one diffusion MRI scan (TR = 2400 ms, TE = 85 ms, flip angle = 90°, 2 mm isotropic voxel resolution, 128 diffusion weighted volumes, b-value = 1500 s/mm$^2$, 9 b = 0 volumes).

The NKI was downloaded in December of 2016 from the INDI S3 Bucket. At the time of download, the dataset consisted of 957 T1w (811 subjects), 914 DWI (771 subjects), and 718 fMRI ("acquisition645"; 634 subjects) images. T1w and DWI images, and tractography results were first filtered based on visual inspection. T1w images were filtered based on artifact, such as ringing or ghosting (43 images) and for FreeSurfer reconstruction failure (105 images) as assesses with the ENIGMA QC tools, leaving 809 T1w images (699 subjects). DWI images were filtered based on corrupt data (13 images) and artifact on fitted fractional anisotropy maps (18 images), leaving 883 images (747 subjects). Tractography was run on 781 images (677 subjects) that had both quality controlled T1w and DWI images. Tractography results were filtered based on artifact, which include failure to resolve callosal, cingulum, and/or corticospinal streamlines or errors resulting in visually sparse streamline densities, resulting in 764 tractography runs (661 subjects). T1w, DWI, and fMRI images were then filtered using computed image quality metrics[104–106]. T1w images were excluded if the scan was marked as an outlier (1.5x the inter-quartile range in the adverse direction) in three or more of following quality metric distributions: coefficient of joint variation, contrast-to-noise ratio, signal-to-noise ratio, Dietrich's SNR, FBER, and EFC. DWI images were excluded if the percent of signal outliers, determined by eddy_qc, was greater than 15%. Furthermore, DWI were excluded if the scan was marked as an outlier (1.5x the inter-quartile range in the adverse direction) in two or more of following quality metric distributions: temporal signal-to-noise ratio, mean voxel intensity outlier count, or max voxel intensity outlier count. fMRI images were excluded if greater than 15% of time frames exceeded 0.5mm framewise displacement. Furthermore, fMRI images were excluded the scan was marked as an outlier (1.5x the inter-quartile range in the adverse direction) in 3 or more of the following quality metric distributions: DVARS standard deviation, DVARS voxel-wise standard deviation, temporal signal-to-noise ratio, framewise displacement mean, AFNI's outlier ratio, and AFNI's quality index. This image quality metric filtering excluded zero T1w images, 16 DWI images, and 21 fMRI images. Following this visual and image quality metric filtering, 809 T1w images (699 subjects), 728 DWI images (619 subjects), and 697 fMRI images (633 subjects). The intersection of subjects with at least one valid T1w, DWI, and fMRI images totaled 567 subjects. Finally, age metadata was available for 542 of these subjects.

T1-weighted images were submitted to FreeSurfer's cortical reconstruction workflow (version 6.0). The FreeSurfer results were used to skull strip the T1w, which was subsequently aligned to MNI space with 6 degrees of freedom. fMRI preprocessing was performed using the fMRIPrep version 1.1.8[107]. The following description of fMRI preprocessing is based on fMRIPrep's documentation. This workflow utilizes ANTs (2.1.0), FSL (5.0.9), AFNI (16.2.07), FreeSurfer (6.0.1), nipype[108], and nilearn[109]. Each T1w was corrected using `N4BiasFieldCorrection`[96] and skull-stripped using `antsBrainExtraction.sh` (using the OASIS template). The ANTs-derived brain mask was refined with a custom variation of the method to reconcile ANTs-derived and FreeSurfer-derived segmentations of the cortical gray-matter of Mindboggle[110]. Brain tissue segmentation of cerebrospinal fluid (CSF), white-matter (WM) and gray-matter (GM) was performed on the brain-extracted T1w using `fast`[99]. Functional data was slice time corrected using `3dTshift` from AFNI and motion corrected using FSL's `mcflirt`. "Fieldmap-less" distortion correction was performed by co-registering the functional image to the same-subject T1w with intensity inverted[111] constrained with an average fieldmap template[112], implemented with `antsRegistration`. This was followed by co-registration to the corresponding T1w using boundary-based registration[113] with 9 degrees of freedom, using `bbregister`. Motion correcting transformations, field distortion correcting warp, and BOLD-to-T1w transformation warp were concatenated and applied in a single step using `antsApplyTransforms` using Lanczos interpolation. Frame-wise displacement[114] was calculated for each functional run using the implementation of Nipype. The first four frames of the BOLD data in the T1w space were discarded. Diffusion images were preprocessed following the "DESIGNER" pipeline using MRTrix (3.0)[115,116], which includes denoising, Gibbs ringing and Rician bias correction, distortion and eddy current correction[117] and B1 field correction. DWI were then aligned to their corresponding T1w and the MNI space in one interpolation step with B-vectors rotated accordingly. Local models of white matter orientation were estimated in a recursive manner[118] using constrained spherical deconvolution[98] with a spherical harmonics order of 8. Tractography was performed using Dipy's `Local Tracking` module[97]. Probabilistic streamline tractography was seeded five times in each white matter voxel. Streamlines were propagated with a 0.5 mm step size and a maximum turning angle set to 20°. Streamlines were retained if longer than 10 mm and with valid endpoints, following Dipy's implementation of anatomically constrained tractography[101]. Streamline count between regions-of-interest were normalized by dividing the count between regions by the geometric average volume of the regions.

**Estimating group-representative structural connectivity network.** The output of the tractography algorithm generated subject-level estimates of streamlines for both the NKI and HCP datasets. In general, subjects' connectomes are variable. A fraction of this variability reflects true individual differences, while another fraction reflects unwanted noise, e.g. random variation. One strategy for reducing noise is to aggregate data from many individuals to construct a group-representative consensus connectome. Here, we follow[30] and generate distant-dependent connectomes for both the NKI and HCP datasets. Briefly, this procedure bins edges by their length and, within each distance bin, identifies the edges that are most consistently present across the full set of subjects. Compared to standard approaches, which retains the most consistent edges irrespective of their length, consensus networks generated using this procedure are more representative of single-subject connectomes–i.e. has more properties in common. Note that this distance-preserving consensus procedure is applied separately to within- and between-hemisphere edges. Note also that for the NKI dataset, the consensus connectome was constructed using data from subjects aged 18-35 years. Finally, we made the decision to use the same (but dataset-specific) group representative for all HCP and NKI subjects. The rationale behind this decision was that it allowed us to discount the possibility that differences in model performance–e.g. fitness or edge weights–was driven by differences in the structural connectivity.

## Dataset: mouse anatomical and functional connectivity

**Mouse resting state fMRI data.** All in vivo experiments were conducted in accordance with the Italian law (DL 2006/2014, EU 63/2010, Ministero della Sanitá, Roma) and the recommendations in the Guide for the Care and Use of Laboratory Animals of the National Institutes of Health. Animal research protocols were reviewed and consented by the animal care committee of the Italian Institute of Technology and Italian Ministry of Health. The rsfMRI dataset used in this work consists of $n = 19$ scans in adult male C57BL/6J mice that are publicly available[119,120]. Animal preparation, image data acquisition, and image data preprocessing for rsfMRI data have been described in greater detail elsewhere[120]. Briefly, rsfMRI data were acquired on a 7.0-T scanner (Bruker BioSpin, Ettlingen) equipped with BGA-9 gradient set, using a 72-mm birdcage transmit coil, and a four-channel solenoid coil for signal reception. Single-shot BOLD echo planar imaging time series were acquired using an echo planar imaging sequence with the following parameters: repetition time/echo time, 1000/15 ms; flip angle, 30°; matrix, 100 × 100; field of view, 2 × 2 cm²; 18 coronal slices; slice thickness, 0.50 mm; 500 ($n = 21$) or 1500 ($n = 19$) volumes; and a total rsfMRI acquisition time of 30 min.

Image preprocessing has been previously described in greater detail elsewhere[120]. Briefly, timeseries were despiked, motion corrected, skull stripped and spatially registered to an in-house EPI-based mouse brain template. Denoising and motion correction strategies involved the regression of mean ventricular signal plus 6 motion parameters. The resulting time series were band-pass filtered (0.01–0.1 Hz band) and then spatially smoothed with a Gaussian kernel of 0.5 mm full width at half maximum. After preprocessing, mean regional time-series were extracted for 182 regions of interest (ROIs) derived from a predefined anatomical parcellation of the Allen Brain Institute (ABI[1,121]).

**Mouse anatomical connectivity data.** The mouse anatomical connectivity data used in this work were derived from a voxel-scale model of the mouse connectome made available by the Allen Brain Institute[122,123] (https://data.mendeley.com/datasets/dxtzpvv83k/2).

Briefly, the mouse structural connectome was obtained from imaging-enhanced green fluorescent protein (eGFP)-labeled axonal projections derived 428 viral microinjection experiments, and registered to a common coordinate space[1]. Under the assumption that structural connectivity varies smoothly across major brain divisions, the connectivity at each voxel was modeled as a radial basis kernel-weighted average of the projection patterns of nearby injections[123]. Following the procedure outlined in[122], we re-parcellated the voxel scale model in the same 182 nodes used for the resting-state fMRI data, and we adopted normalized connection density (NCD) for defining connectome edges, as this normalization has been shown to be less affected by regional volume than other absolute and/or relative measure of interregional connectivity[124].

## Fitting edge weights

Here, we use a regression-based framework for assigning weights to existing structural connections. Our approach is simple; we assume that at time $t$ the state of region $i$ (level of fMRI BOLD activity) is a function of its neighbors' states at time $t - 1$ plus an offset (bias). That is:

$$y_i(t) = \sum_{j \in \Gamma_i, j \neq i} W_{ji} y_j(t-1) + c_i. \tag{2}$$

Here, $y_i(t)$ refers to the level of activity in region $i$ at time $t$, $\Gamma_i$ is the set of $i$'s connected neighbors (their indices). We use linear regression and ordinary least squares to estimate the parameters $W_{ji}$ and $c_i$ separately for each node $i$. Thus, the resulting matrix $W \in \mathbb{R}^{n \times n}$, is sparse and preserves *exactly* the binary structure of the white-matter connectivity. However, the weights can take on both positive and negative valence. The resulting network is also asymmetric–i.e. in general, the $W_{ij} \neq W_{ji}$.

## Null models

We fit the linear model using data pooled from all participants and scans. The model fitness was quantified as the mean squared error (MSE) of the observed activity time series and the time series predicted by the model. We compared the empirical msE against five null models.

- Minimally wired null model: Generates a synthetic structural network comprised of the $m$ least costly connections, where $m$ is the same number of connections as the observed network. Because this network contains only short-range (low cost) connections, this null model assesses how long-distance connections contribute to model fitness.
- *Re-ordered null model:* Randomly permutes node order, effectively endowing nodes with a different number and set of neighbors than they have in the original network. This model assesses the contributions of specific neighbors to model fitness.
- "Spin" null model: Randomly permutes node order while approximately preserving inter-regional Euclidean distances. This model can be viewed as a constrained versions of the re-ordered null model, in that it only allows particular subset of permutations.
- Topological null model: In this model, each node makes the same number of connections as in the original network. However, those connections, which define nodes' neighborhoods, are formed at random. This model assesses whether networks with identical degree distribution yield similar fitness values as the original network.
- Temporal null model: For each scan, parcel time series are circularly shifted by some random integer. This procedure preserves temporally invariant properties of each time series, like their mean and standard deviation, and approximately preserves other properties, e.g. power spectrum. However, it destroys inter-regional correlations. In effect, this model tests whether time series with similar statistical properties but no correlation structure could yield comparable fitness values as the original time series.

## Modularity maximization

Here, we used modularity maximization to detect clusters (modules) in brain network data[125,126]. Generically, modularity maximization works by assigning nodes to non-overlapping clusters so that the within-cluster weight of connections maximally exceeds that of a null model. This intuition is formalized by the modularity quality function:

$$Q(\gamma) = \sum_{ij} [W_{ij} - \gamma P_{ij}] \delta(z_i, z_j). \tag{3}$$

In this equation, $W_{ij}$ and $P_{ij}$ are the observed and expected weight of the connection between nodes $i$ and $j$, $z_i \in \{1, ..., K\}$ is a categorical variable that indicates the community to which node $i$ was assigned, $\gamma$ is the structural resolution parameter, and $\delta(z_i, z_j)$ is the Kronecker delta function, which evaluates to 1 when $z_i = z_j$ and 0 otherwise. In short, modularity maximization seeks to optimize the quantity $Q(\gamma)$ by selecting the values of $z_i$.

The modularity maximization framework is general and can test different null hypotheses (null connectivity models) by varying the entries of $P$, the matrix of expected connections and their weights. Here, we test two different null models. The first was proposed in[37] and is designed, specifically, to work with signed connectivity matrices. Under this model, the modularity equation is:

$$Q^*(\gamma) = \frac{1}{k^+} \sum_{ij} [W_{ij}^+ - \gamma P_{ij}^+] \delta(z_i, z_j) - \frac{1}{k^+ + k^-} \sum_{ij} [W_{ij}^- - \gamma P_{ij}^-] \delta(z_i, z_j). \tag{4}$$

Here, the modularity equation includes separate terms for the positive and negative connections. The positive term is weighted more than the negative term (note the scale factors before the summation). This allows modules to be detected in networks with signed connections. However, if this same version of modularity maximization is applied to a network with positive links only, the second term in the equation evaluates to zero and returns the standard modularity equation. Note that in this equation, $P_{ij}^{\pm} = \frac{k_i^{\pm} k_j^{\pm}}{2m^{\pm}}$.

Here, we use this quality function in two ways. In the main text, we optimize $Q^*$ 1000 times for both the asymmetric, weighted, and signed network as well as the fiber density network. These results are shown in Fig. 2. In the supplement, we combine this quality function with a hierarchical consensus algorithm[127], in which we first vary the values of $\gamma$ over all possible ranges to obtain a representative sample of communities (1,000,000 repetitions in total), and second, use these samples to construct a hierarchical dendrogram that organizes the noisy individual samples into hierarchically related consensus communities. The results of this analysis are shown in Fig. S9.

We also used a second version of the modularity equation that was originally proposed for analysis of physical systems[128]. Briefly, the equation reads:

$$Q = \sum_{ij}[W_{ij} - \langle W \rangle A_{ij}]\delta(z_i, z_j). \tag{5}$$

Here, the matrix $A$ is the binary matrix of connections that exist in the empirical and weighted connectivity matrix. $\langle W \rangle$ is the mean weight of existing connections. In other words, this modularity equation preserves the topology of the network, but assumes that edge weights are assigned randomly and uniformly. The results of this analysis are presented in Fig. S11. We note that, in principle, a resolution parameter could be incorporated into this formulation of the modularity quality function as well by replacing $\langle W \rangle$ with a tunable $\gamma$ parameter.

## Network statistics

In addition to modularity, we calculated several other network metrics. These include efficiency, characteristic path length, signed strength, and signed clustering coefficient. In this section we define those measures in detail.

- *Shortest paths*. Both efficiency and characteristic path length are defined based on a network's shortest path structure. Consider source and target nodes, $s$ and $t$. The shortest path from the $s$ to $t$ can be estimated easily using the Floyd-Warshall algorithm. In weighted networks where edge weights as interpreted as measures of affinities it requires that the user first map those weights to measures of cost. For networks with positive connections only, a straightforward way to do this is to transform $C_{ij} = W_{ij}^{-\gamma}$, where the most common value for the parameter is $\gamma = 1$. For signed networks, like the ones used here, we use the same transformation, but only after we add an offset to each edge so that all weights are greater than zero. Our strategy for doing so involved first subtract the smallest (most negative) edge weight from the network. This ensures that all edges have a weight greater than zero, except for the single edge corresponding to the most negative weight, which has a cost of 0. We then add to every edge an even smaller offset–in this case the weakest edge weight in the fiber density matrix. This guarantees that all pairs of nodes connected by a fiber tract have nonzero weights. Once a network's affinity-based weights have been transformed to costs, algorithms like the Floyd-Warshall algorithm find the shortest-i.e. least costly–path between all pairs of nodes. This algorithm returns two outputs: 1) the total cost incurred by following said path and 2) the number of steps (hops) along said path. Here, we use the hop data but not that, in principle, one could repeat all subsequent analyses

using the cost data, instead. Let $H_{st}$ be the number of hops on the shortest path from the source $s$ to the target $t$.

- *Characteristic path length*. The characteristic path length of this network is calculated as:

$$L = \frac{1}{n(n-1)} \sum_{i,j \neq i} H_{st}. \tag{6}$$

- *Efficiency*. The efficiency of this network is calculated as:

$$E = \frac{1}{n(n-1)} \sum_{i,j \neq i} \frac{1}{H_{st}}. \tag{7}$$

- *Clustering coefficient*. The local clustering coefficient is calculated for each node $i$. Intuitively, it measures the extent to which node $i$'s neighbors are also connected to one another. It can be calculated easily for each node as the density of the subgraph composed of those neighbors. Here, we calculate clustering coefficients for each node in the network based on their positive connections and negative connections, separately. The values reported in the main text ignore the actual weight but preserve sign.

- *Strength*. Node strength – or weighted degree – the total weight of connections incident upon node $i$. For an undirected network, it is calculated as: $s_i = \sum_j W_{ij}$. For a directed network, we calculate strength as the average of a nodes' incoming and outgoing connections, i.e. $s_i = \frac{\sum_j W_{ij} + \sum_j W_{ji}}{2}$. Here, we also differentiate between a node's positive and negative strength. Let $W^+$ and $W^-$ be the networks of positive and negative connections only. For the network of negative connections, we conveniently flip the sign of each connection. Then we calculate each nodes' signed strength as $s_i^{\pm} = \frac{\sum_j W_{ij}^{\pm} + \sum_j W_{ji}^{\pm}}{2}$.

- *Partition laterality*. We calculated partition laterality following[39]. For a given community $c$, we calculate its uncorrected laterality as $\Lambda_c = \frac{|N_r - N_l|}{N_c}$. Here, $N_c$ is the number of nodes in $c$ and $N_r$ and $N_l$ are the number of those nodes in the right and left hemispheres, respectively. When the community has a balanced number of nodes from both hemispheres its laterality is close to zero; if it is left- or right-dominant, then the value is close to 1. For a partition comprised of communities $c_1, ..., c_K$, we calculate the partition laterality as $\Lambda = \frac{1}{N}(\sum_c N_c \Lambda_c - \langle \sum_c N_c \Lambda_c \rangle)$. Here, the term $\langle \sum_c N_c \Lambda_c \rangle$ indicates the expected laterality under a null model in which nodes get randomly assigned to one hemisphere or another. Note that here we cannot use spin tests for the permutation; the spin tests preserves hemisphere labels and a "spun" partition would have laterality exactly equal to that of the original, unpermuted partition.

## Neural mass models

Many studies have tried to link brain structure and function[35]. One popular strategy for doing so is to use an estimate of anatomical connectivity to generate synthetic covariance matrices (either directly or by first generating synthetic neural time series and calculating their covariance empirically). The synthetic covariance matrices can then be compared to the empirical FC, usually as a correlation of their edge weights. The resulting coefficient serves as a measure of structure-function coupling. Here, we analyzed two models for generating synthetic covariance matrices or time series based on population-level "neural mass" models (NMMs).

- *Galán model*. We follow work by[63] for estimating the inter-areal covariance matrix, **C**, based on a linearization of Wilson-Cowan

dynamics for neuronal populations[129]. The element $C_{ij} \in \mathbf{C}$ denotes the covariance of activity in area $i$ with that of area $j$. In more detail, we let $\mathbf{u}(t) = \{u_1(t), ..., u_N(t)\}$ be the vector of brain areas' states (activity levels) at time $t$. Under these dynamics, brain areas' states evolve as:

$$\mathbf{u}(t + \Delta t) = \mathbf{A}\mathbf{u}(t) + \xi(t), \tag{8}$$

where $\xi(t)$ is uncorrelated Gaussian noise and $\Delta t$ is a single time step. Here, the generalized coupling matrix, $\mathbf{A}$, is based on the structural connectivity matrix, $\mathbf{W}$, and was defined as:

$$\mathbf{A} = (1 - \alpha \Delta t)\mathbf{I} + \mathbf{W}\Delta t, \tag{9}$$

where $\alpha$ is a leak variable within each brain area and $\mathbf{I}$ is the identity matrix. As in[63], we fixed $\alpha = 2$. Conveniently[129], showed that brain areas' pairwise covariances (summarized by the matrix $\mathbf{C} \in \mathbb{R}^{N \times N}$) can be estimated directly from the spectral properties of $\mathbf{A}$ and the covariance of the noise terms $\xi(t)$. As with covariance matrices estimated from recorded time series of brain activity, we interpret $\mathbf{C}$ as an estimate of functional connectivity. See[129] for more details.

- *Reduced Wong-Wang mean field model.* We also studied a second biophysical model for fMRI BOLD data. Unlike the Galán model, which calculates the covariance structure analytically given a structural connectivity matrix, this model generates simulated time series, first by using a reduced spiking neural network to generate population-level time courses, and second by convolving these data with a hemodynamic model. The spiking network model evolves according to the following differential equations:

$$\dot{S}_i = -\frac{S_i}{\tau_S} + r(1 - S_i)H(x_i) + \sigma v_i(t)$$
$$H(x_i) = \frac{ax_i - b}{1 - exp(-d(ax_i - b))} \tag{10}$$
$$x_i = wJS_i + GJ\sum_j W_{ij}S_j + I$$

In this equation, $x_i$, $H(x_i)$, and $S_i$ are the total input current, population firing rate, and synaptic gating for region $i$. The input current, $x_i$ depends on recurrent connection strength, $w$, excitatory input, $I$, and inter-regional information "flow", which is calculated as the sum of region $i$'s connected neighbors' synaptic gating, weighted by the global coupling constant, $G$, and synaptic coupling constant, $J$. Following[130], we set the parameters of the input-output function, $H(x_i)$ to $a = 270$ n/C, $b = 108$ Hz, and $d = 0.154$ s. Kinetic parameters for synaptic activity were fixed at $r = 0.641$ and $\tau_s = 0.1$ s. The variable $v_i(t)$ is uncorrelated Gaussian-distributed noise whose variance is scaled by $\sigma$. This model generates neural activity at sub-millisecond timescales. Again, following[130], population level activity is input to the Balloon-Windkessel hemodynamic model[131], which yields simulated fMRI BOLD time courses for every brain region.

## Reporting summary
Further information on research design is available in the Nature Portfolio Reporting Summary linked to this article.

## Data availability
All of the human neuroimaging datasets used here are publicly available. Data from both of the Human Connectome Project datasets can be downloaded at: https://db.humanconnectome.org/. Information on accessing the NKI dataset can be found at: http://fcon_1000.projects.nitrc.org/indi/enhanced/neurodata.html. All data generated and used in the main figures of this study are provided in the Supplementary Information/Source Data file. Source data are provided with this paper.

## Code availability
Code to produce an asymmetric, weighted and signed connectome with functional and structural data, and to replicate many of our results can be found at: https://github.com/JacobColbyTanner/asymmetric_weighted_and_signed_connectome-main DOI for the toolbox: https://doi.org/10.5281/zenodo.11036029.

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

## Acknowledgements

A.S.T. acknowledges support by FCT – Fundação para a Ciência e Tecnologia – through the LASIGE Research Unit, ref. UIDB/00408/2020 (https://doi.org/10.54499/UIDB/00408/2020) and ref. UIDP/00408/2020 (10.54499/UIDP/00408/2020).

## Author contributions

J.C.T. and R.F.B. conceived the study, processed data, carried out all analyses, wrote, edited, and revised the submitted manuscript. J.F. processed all of the human MRI data. L.C. and A.G. collected and processed all of the mouse data. B.M. provided brain map data. A.S.T. and C.S. advised on analyses. All authors edited the manuscript.

## Competing interests

The authors declare no competing interests.
