## [Peer Review File · Nature Communications]

nature portfolio

Peer Review File

A multi-modal, asymmetric, weighted and signed description
of anatomical connectivityReviewer #1 (Remarks to the Author):

The manuscript from Tanner et al. describes a framework for defining connectome data in the context of multi-modal approaches using structural and functional MRI as well as DTI. The paper is very well-written, thorough, and would be of interest to the network neuroscience community. The figures are nicely laid out and relatively easy to follow. The authors should be commended on a really nice manuscript.

That being said, I have a couple of concerns that I believe are worth addressing, but aren't related to any findings or constructs presented by the authors. Instead, I found the manuscript to read much more as a resource as opposed to a set of findings that advances our understanding of a specific line of research/scientific questioning. As such, I think the paper can increase its significance to the scientific community by ensuring that the framework the authors have developed contains tools that are easily usable by other researchers with a similar focus. The authors have taken advantage of data sharing efforts, but without software that is easily disseminable, true reproducibility is difficult for the majority of labs. If the paper is deemed worthy of publication here, I would push the authors to develop tools for the broad community to use; my recommendation would be contingent on that.

Otherwise, as I've said, this is a very nice paper that will be an excellent resource for the field.

Reviewer #2 (Remarks to the Author):

The paper presents an approach that uses multi-modal data to estimate directed weights of brain connections and provide an alternative to current weighting schemes. Weights for each region are obtained using regression models that describe functional activity. The paper demonstrates that such a connectome can be used to predict fMRI-observed activity at an individual level, while showing anticipated network properties (bilaterally symmetric, modular structure and a shortest-path backbone). Further experiments show how these weights change across the lifespan and during movie watching. This is a novel approach and the manuscript reflects a lot of interesting work, for which I would like to applaud the authors. I do have some reservations though and I feel addressing those would benefit the paper and its readers.

The proposed approach uses functional data to estimate brain connections. Hence it falls into the category of inferring "structure from function" approaches, but -in addition to existing similar approaches that use only functional data- an anatomical connectivity backbone is used to effectively constrain this estimation. This is a perfectly valid approach, but my objection is that the authors call this a redefinition of "anatomical/structural connectivity" (from title to Intro, Results, Discussion etc), which to me is inherently linked into estimating properties of white matter. I hence found this quite confusing and counter-intuitive throughout the paper and I believe this aspect needs to be reconsidered in terms of how it is presented. The paper will also benefit by comparisons with other approaches that attempt the same connectivity inference from functional data, e.g. partial correlation of resting-state data. Specifically:

- A group-averaged binary structural connectivity (SC) matrix is used, even for deriving subject-specific weighted matrices. It is claimed that this is done in order to **avoid** differences in anatomical connectivity influencing the results and to ensure that these between-subject differences are driven by resting-state dynamics (beginning of p.4). This goes against the story of the paper until that point, even the paper title itself, where a redefined description of "anatomical connectivity" is promised. The approach followed is to intentionally avoid capturing differences in anatomical connectivity and instead define weights of a group-average binarised matrix, based on fMRI data. I have no issue with doing that, but I struggle to understand why this is called a redefinition of the anatomical connectome, if anatomical connections are not the focus nor what is estimated. I believe this is unnecessarily confusing.

- Confusion continues when the regression approach is applied to mouse chemical tracing data, which are, by definition, directed (Fig S13). Then tracer weights and their directionality (that have a very clear neuroanatomical meaning in terms of traced axons) are removed, the matrix is

binarized and new weights and directionality are inferred from fMRI data of different animals.... I struggle to follow the argument that this as a redefinition of anatomical connectivity. What this new approach really offers is an anatomically-constrained version of functional connectivity (FC), rather than a redefinition of SC. In support of this, Figure S1 suggests that (somewhat unsurprisingly) the two closest to the estimated weights are partial and full correlation rather than tractography weights (or any other dMRI-derived metric). Hence, similarly to comparing against tractography-derived weights, I would suggest that the more relevant comparisons are against approaches which are inherently closer to the proposed one, for instance partial correlation of fMRI timeseries.

- Directed weights: Do these really capture directionality in anatomical connectivity? Incoming vs Outgoing connections as claimed? Or not? If they do, shouldn't these directionalities agree with the directionality in the tracer data? In the relevant section of the paper, where asymmetry on connection weights is explored, I feel there is a missed opportunity to answer this question. The authors do a good job characterising where weight asymmetries occur, but they do not answer what these asymmetries really reflect. Given the analysis they already did against the tracer data in the mouse (Fig. S13), and given that tracer weights are directed, comparison of their inferred directionality against observed directionality with tracers should be relatively straightforward and would be very informative on what "directionality" represents in the proposed approach.

- One of the major conceptual limitations when estimating SC from dMRI is that dMRI-tractography follows water and not necessarily axons, hence these matrices can have many false positives. The proposed approach seems to ignore this fact and subsequently takes an SC matrix for granted, binarise it and learn its weights from a different modality. I think it would be great to get a feel of how false positive connections in the binarised graph are treated in the weighted graph. For instance, do known false positives get assigned small weights after regression with the fMRI data? Or are false negatives upscaled? Hence the weighting can be used to somehow "filter" the original matrix?

Or the approach simply propagates/amplifies false positives, in which case interpretation issues and challenges remain unresolved? One would hope that a false positive connection (that reflects dMRI limitations) won't be important in predicting fMRI data (that are independent to dMRI and their limitations). Such type of indirect "filtering" of SC using multi-modal data would be a genuine contribution to the field and would be worth exploring with the proposed framework in my opinion. And would considerably support the claim about redefining SC.

- Along the same lines, in Fig. 1. explorations are presented on testing how stable the regressed weights are when subsampling the fMRI data. But what is not tested is how stable these weights are to perturbations of the original SC matrix. An SC matrix can be obtained in a number of ways, binarising it also can be done in a number of ways. How sensitive or not the results are to how this SC matrix is obtained? Particularly given that it contains errors?

Minor comments

- The paper is quite long. Some sections can be shortened by moving whole paragraphs to supplementary and summarise in a sentence or two in main text (e.g. page 5, paragraph around Figures S10, S11, offers extra evidence, but nothing new in terms of the main message of the section, so this can all go into supplement. Similarly for other sections. In Methods for instance there are very long sections describing data and processing that are publicly available, I would focus these sections on the extra processing you did.)

- Fig. 4b, colour code should start from zero, as absolute values are shown

- I would make clear early on whether by BOLD activity the authors refer to resting-state or task.

Dear Reviewers,

We thank you for their commitment to helping us improve this submission. In responding to your comments, we believe that our enclosed submission is much improved.

Throughout this response letter we refer to comments made by the reviewers in black text. Our responses are shown in **blue text**. Changes to the manuscript are depicted in *italicized red text* to highlight the specific edits. Reviewer 2 also asked that we condense the manuscript. This involved replacing many paragraphs with shorter versions. Deleted passages are written in *italicized green text*.

When appropriate, we note the section name where changes were implemented. In addition to this response letter, we include two versions of the revised manuscript: a “marked” version that tracks all of the changes made since the original submission and a “clean” version that includes all changes but without any highlights.

Jacob Tanner

Rick Betzel

REVIEWER COMMENTS

Reviewer #1 (Remarks to the Author):

The manuscript from Tanner et al. describes a framework for defining connectome data in the context of multi-modal approaches using structural and functional MRI as well as DTI. The paper is very well-written, thorough, and would be of interest to the network neuroscience community. The figures are nicely laid out and relatively easy to follow. The authors should be commended on a really nice manuscript.

We sincerely thank reviewer for their thoughtful and positive evaluation of our manuscript.

R1. Comment #1:

That being said, I have a couple of concerns that I believe are worth addressing, but aren't related to any findings or constructs presented by the authors. Instead, I found the manuscript to read much more as a resource as opposed to a set of findings that advances our understanding of a specific line of research/scientific questioning. As such, I think the paper can increase its significance to the scientific community by ensuring that the framework the authors have developed contains tools that are easily usable by other researchers with a similar focus. The authors have taken advantage of data sharing efforts, but without software that is easily disseminable, true reproducibility is difficult for the majority of labs. If the paper is deemed worthy of publication here, I would push the authors to develop tools for the broad community to use; my recommendation would be contingent on that.

Otherwise, as I've said, this is a very nice paper that will be an excellent resource for the field.

We would like to sincerely thank the reviewer for their thoughtful review of our manuscript. We agree with the reviewer that a clearly documented repository would, indeed, be a useful resource for the field. For this reason, we created a github repository that includes functions that implement the analyses reported in the manuscript. They include functions to reweight structural connections with functional imaging data using our model (the principal contribution of our submission), alongside other functions for investigating weight asymmetries, module lateralization, edge usage and more.

Please see the link below for our repository as well as a copy of the README that describes each of the included functions:

This toolbox presents a set of functions that can be used to create and analyze asymmetric, weighted and signed anatomical networks as presented in the paper: Tanner, J., Faskowitz, J., Teixeira, A. S., Seguin, C., Coletta, L., Gozzi, A., ... & Betzel, R. (2022). Reweighting the connectome: A multi-modal, asymmetric, weighted, and signed description of anatomical connectivity.

Constructing such a network requires two separate inputs:

1. A binarized structural brain network (directed or undirected) describing the anatomical connections between all nodes in the network. This network should be in the form of an adjacency matrix.
2. Brain activity time series with the same number of nodes as the binarized structural brain network. This time series data should be organized into a matrix where the rows represent time points and the columns represent node activity (time x node).

The *main.m* file uses some example data from a de-identified subject from Human Connectome Project dataset in order to create and plot an asymmetric, weighted and signed network from (1) a binarized structural brain network, and (2) a time series of brain activity. Additionally, this code will also call a variety of functions to analyze this new network and compare it with a fiber density weighted network, the results of which are plotted.

The functions included in this toolbox are listed below. More detailed comments and instructions for usage can be found within each function.

1. *fcn_fit_model* : takes in a binary structural network and brain activity time series and returns an asymmetric, weighted and signed connectome, as well as various performance metrics.
2. *fcn_get_asymmetry* : returns three measures of asymmetry. The asymmetry matrix where each entry is the difference between the weight of edge(i,j) and edge(j,i), the absolute asymmetry matrix where each entry is the absolute difference between edge(i,j) and edge(j,i), and the sign asymmetry matrix that describes where edge(i,j) and edge(j,i) had different signs.
3. *fcn_get_in_out_similarity* : returns an array of values (one for each node in the network) that describe the similarity of the in-weights and out-weights for each node. The in-weights are the weights that were used by a linear regression model to predict the future activity of this node. So, the in-weights are weights describing when other nodes are used to predict this nodes activity. In contrast, the out-weights are when this nodes activity is used to predict a different nodes activity.
4. *fcn_consensus_communities* : takes in multiple partitions of the same connectivity matrix into communities/modules and returns a consensus partition. This function includes the use of two functions to relabel and identify unique partitions (*fcn_relabel_partitions* and *fcn_unique_partitions* respectively).
5. *fcn_sort_communities* : relabels the communities of a partition such that the largest community gets labeled 1, the next largest gets labeled 2, and so on.
6. *fcn_get_geometric_null* : creates a null connectivity matrix for usage in modularity maximization. This null assigns every existing edge in the network to the mean weight value across all existing edges.
7. *fcn_bilaterality* : takes in a set of community labels as well as labels for the hemisphere each node belongs to and computes a measure of laterality. This measure describes how much each of the modules tends to be concentrated in one of the two hemispheres.
8. *fcn_get_edge_usage* : counts the number of times each edge is used in the shortest paths between all nodes in the network.

In addition, this toolbox uses a number of functions from the Brain Connectivity toolbox which is publicly available at: <https://sites.google.com/site/bctnet/>.

If you use this toolbox in your published work, please cite our paper:

*Tanner, Jacob, Joshua Faskowitz, Andreia Sofia Teixeira, Caio Seguin, Ludovico Coletta, Alessandro Gozzi, Bratislav Mišić, and Richard F. Betzel. "Redefining the connectome: A multi-modal, asymmetric, weighted, and signed description of anatomical connectivity." *bioRxiv* (2022): 2022-12. <https://doi.org/10.1101/2022.12.19.519033>.*

The link to the repository is below

https://github.com/JacobColbyTanner/asymmetric_weighted_and_signed_connectome-main

We also include a link to the repository in the main text under the section “**Code Availability**”:
The new text reads:

*“Code to produce an asymmetric, weighted and signed connectome with functional and structural data, and to replicate many of our results can be found at:
https://github.com/JacobColbyTanner/asymmetric_weighted_and_signed_connectome-main.git”*

Reviewer #2 (Remarks to the Author):

The paper presents an approach that uses multi-modal data to estimate directed weights of brain connections and provide an alternative to current weighting schemes. Weights for each region are obtained using regression models that describe functional activity. The paper demonstrates that such a connectome can be used to predict fMRI-observed activity at an individual level, while showing anticipated network properties (bilaterally symmetric, modular structure and a shortest-path backbone). Further experiments show how these weights change across the lifespan and during movie watching. This is a novel approach and the manuscript reflects a lot of interesting work, for which I would like to applaud the authors. I do have some reservations though and I feel addressing those would benefit the paper and its readers.

We sincerely appreciate the reviewer's thoughtful and detailed review of our manuscript. These insights and suggestions have been invaluable in enhancing the quality and clarity of our work. In what follows, we describe the edits made and additional analyses carried out to address the reviewer's concerns.

R2. Comment #1:

The proposed approach uses functional data to estimate brain connections. Hence it falls into the category of inferring "structure from function" approaches, but -in addition to existing similar approaches that use only functional data- an anatomical connectivity backbone is used to effectively constrain this estimation. This is a perfectly valid approach, but my objection is that the authors call this a redefinition of "anatomical/structural connectivity" (from title to Intro, Results, Discussion etc), which to me is inherently linked into estimating properties of white matter. I hence found this quite confusing and counter-intuitive throughout the paper and I believe this aspect needs to be reconsidered in terms of how it is presented. The paper will also benefit by comparisons with other approaches that attempt the same connectivity inference from functional data, e.g. partial correlation of resting-state data. Specifically:

- A group-averaged binary structural connectivity (SC) matrix is used, even for deriving subject-specific weighted matrices. It is claimed that this is done in order to *avoid* differences in anatomical connectivity influencing the results and to ensure that these between-subject differences are driven by resting-state dynamics (beginning of p.4). This goes against the story of the paper until that point, even the paper title itself, where a redefined description of "anatomical connectivity" is promised. The approach followed is to intentionally avoid capturing differences in anatomical connectivity and instead define weights of a group-average binarised matrix, based on fMRI data. I have no issue with doing that, but I struggle to understand why this is called a redefinition of the anatomical connectome, if anatomical connections are not the focus nor what is estimated. I believe this is unnecessarily confusing.

We thank the reviewer for these useful suggestions. We agree that the phrasing in our paper may inadvertently imply an attempt to infer structural connections from functional data. However, this is not the goal of our paper. Rather, our primary aim is to reweight or otherwise "inform" structural connections with functional imaging data. Thus, we can use our approach to address the question: "How can existing binary structural networks be reinterpreted and enhanced by

incorporating elements of functional imaging data?” To more accurately reflect this aim, we changed our title from: “Redefining the connectome” to “Reweighting the connectome”.

We also agree that there are many strategies for establishing the weights of inter-regional white-matter connections. Many of these measures—e.g. fractional anisotropy, mean diffusivity, streamline count—are interpreted as indirect estimates of interregional communication efficacy. However, these and other measures are generally derived from dMRI data, disallowing functional imaging data to inform their value. We propose integrating functional imaging data into the estimation of structural edge weights. We present evidence that this approach yields weights that are largely uncorrelated with existing dMRI measures, highlight specific applications where this approach yields useful insights, and speculate that there may be other cases where functionally informed structural connectivity weights are useful.

We also note that our approach is readily applicable to participant-level SC (though we contend that, given the relative newness of our approach, benchmarking its properties on a “group-representative” is also appropriate and facilitated the movie-watching and lifespan analyses showcased in the main text).

To illustrate the generalizability of the method to subject-level data, we performed a supplementary analysis wherein we trained subject-specific models using subject-specific resting-state fMRI and subject-specific structural connectivity matrices. We then compared the similarity of the edge weights estimated using these models with the weights estimated using the group-representative matrix. We found that subject-specific weights were similar to those of group model reported in the main text (mean similarity $r = 0.66$; See Fig. S20, copied below for convenience).

We have revised our manuscript to better articulate these points.

Our new title:

“Reweighting the connectome: a multi-modal, asymmetric, weighted and signed description of anatomical connectivity”

Additionally, we added the following text in the beginning of the “**Discussion**” section to clarify that we are not intending to infer structure from function:

“We note that our approach shares some features with existing frameworks--e.g. effective connectivity and dynamic causal [1-6] --in that it returns directed connections. However, our weighting scheme is not generative--i.e. it cannot be used to generate new synthetic time series data. Additionally, our approach is not seeking to solve the inverse problem--i.e. inferring structure from function. Rather, it is explanatory and represents a means of weighting already reconstructed fiber tracts. It is therefore distinct from extant approaches in network neuroscience, and presents opportunities for multiple follow-up studies and applications in other neuroscientific disciplines.”

Please see below for a copy of Figure S20 describing our new subject-specific weight analysis:

FIG. S20. **Subject-specific model weights are more similar within subjects, but also highly similar to group model weights.** (a) Similarity matrix showing the similarity of all subject-specific models. Subject models were trained in the following manner. Four models were trained per subject. The first model used fMRI scan sessions 2,3 & 4. The second model used fMRI scan sessions 1,3 & 4. Following the general pattern of using three out of four of the scans. Along the block diagonal of this similarity matrix you can see high similarity values that correspond to high within-subject similarity. (b) Boxplot showing the within subject similarity versus the between subject weight similarity values from the previous similarity matrix. Within subjects are significantly greater than between subject similarity values (two-sample t -test; $p < 10^{-15}$). Additionally, between subject similarity values were significantly greater than zero (one-sample t -test; $p < 10^{-15}$). Importantly, the structural connectivity matrices of these between subject models were individualized to each subject, so the weight similarity values correspond to the similarity of the overlapping edges in all subjects structural connectivity matrices. (c) Finally, we trained 95 new subject-specific models (all four fMRI scan sessions for each subjects model), and then compared the weights of edges that existed in all subject specific models with the weights of those edges in the group estimated model weights that was used in the main text (total number of overlapping connections was 9132). We found that these weights were highly similar.

We also added new text that references this figure in the **Results** section entitled “**Fitting and benchmarking asymmetric, weighted, and signed structural connectivity**”:

“Brain activity dynamics and its correlation structure are deeply individualized [31, 32]. A good model of brain activity, therefore, must also exhibit subject specificity. To assess whether model performance was, indeed, subject specific, we estimated weights using three of every subject’s four resting state scans, and used those weights to predict the activity of the held-out scan (as well as the activity of all other scans and subjects; Fig. 1g). We found that the error (mean squared error) was lower for the held-out scans than for the scans of any other subjects (two-sample t -test; $p < 10^{-15}$; Fig. 1h). Here, as in subsequent single-subject/-scan analyses, we fit edge weights using the same group-representative connectivity mask. This ensures that any differences between individuals are not driven by differences in the underlying anatomical connectivity, but driven jointly by differences in edge weights and resting brain dynamics. In a supplementary analysis, we also show that the weights of models with subject-specific fMRI as well as subject-specific structural connectivity are more similar within subjects, than between subjects (Fig. S20a,b; two-sample t -test; $p < 10^{-15}$). Importantly, we also found that the subject specific model weights were highly similar to the group estimated model weights (Fig. S20c; mean similarity $r = 0.66$; one-sample t -test; $p < 10^{-15}$).”

R2. Comment #2:

- Confusion continues when the regression approach is applied to mouse chemical tracing data, which are, by definition, directed (Fig S13). Then tracer weights and their directionality (that have a very clear neuroanatomical meaning in terms of traced axons) are removed, the matrix is binarized and new weights and directionality are inferred from fMRI data of different animals.... I struggle to follow the argument that this as a redefinition of anatomical connectivity. What this new approach really offers is an anatomically-constrained version of functional connectivity (FC), rather than a redefinition of SC. In support of this, Figure S1 suggests that (somewhat unsurprisingly) the two closest to the estimated weights are partial and full correlation rather than tractography weights (or any other dMRI-derived metric). Hence, similarly to comparing against tractography-derived weights, I would suggest that the more relevant comparisons are against approaches which are inherently closer to the proposed one, for instance partial correlation of fMRI timeseries.

- Directed weights: Do these really capture directionality in anatomical connectivity? Incoming vs Outgoing connections as claimed? Or not? If they do, shouldn't these directionalities agree with the directionality in the tracer data? In the relevant section of the paper, where asymmetry on connection weights is explored, I feel there is a missed opportunity to answer this question. The authors do a good job characterising where weight asymmetries occur, but they do not answer what these asymmetries really reflect. Given the analysis they already did against the tracer data in the mouse (Fig. S13), and given that tracer weights are directed, comparison of their inferred directionality against observed directionality with tracers should be relatively straightforward and would be very informative on what "directionality" represents in the proposed approach.

We appreciate the reviewer's insightful observations regarding the application of our regression approach to mouse chemical tracing data. We note that our approach does not seek to validate tract-tracing data. Rather, the goal is to reweight the tracts by incorporating functional imaging data. Through this lens, our approach functionally annotates existing tracts, assigning them different weights. Our exploration is not intended to supplant established methods to estimate anatomical connectivity but to provide an alternative lens through which to view and interpret these connections.

Nonetheless, we agree that it would be useful to directly compare the asymmetry of edge weights we estimated with those obtained from the mouse connectome. In general, we found no obvious correlation ($r = -0.01$, $p = 0.98$; see newly added Fig. S17a copied below for convenience), suggesting that the functionally informed edge weights are largely orthogonal with weights derived directly from tract tracing.

To further investigate the importance of weight asymmetries in our model, we performed a series of additional analyses wherein we forced the trained weights of our model to be symmetric, and then assessed the effect of this intervention on model performance. We tested three symmetrizing strategies which we applied to both mouse and human imaging data:

1. Mean of upper/lower triangle: $A(i, j) = \text{mean}(A(i, j), A(j, i))$.
2. Replace lower triangle weights with weights from upper triangle: $A(i, j) = A(j, i)$.

3. Replace upper triangle weights with weights from lower triangle: $A(j, i) = A(i, j)$.

In all cases, we found that model performance degrades when the weights are forced to be symmetric (see Fig. S17b,c below).

We then performed this analysis again on human imaging data using subject-specific SC and fMRI. We found that the effect holds across subjects. That is, model performance degrades when weights are forced to be symmetric (see Fig. S17d; paired-sample t -test; $p < 10^{-15}$). Additionally, in corroboration with our previous result showing that incoming and outgoing weights were least similar in heteromodal brain systems (see Fig. 4g), we found that model prediction of heteromodal brain systems were consistently more negatively affected by forcing symmetry than unimodal brain systems (see Fig. S17e,f).

Finally, we performed an additional analysis wherein we forced model weights to be symmetric *during* training while using gradient descent. This allowed us to directly test whether or not a symmetric model could be trained to perform as well as an asymmetric model. We found that model performance (as measured by mean squared error; MSE) was significantly worse across 95 separately trained symmetric models for unrelated subjects from the Human Connectome Project (see Fig. S17g-i; paired-sample t -test; $p < 10^{-15}$).

Collectively, these supplementary results suggest that, while edge weights estimated from our model diverge from traditional tracer-based assessments, they nonetheless offer unique insight into connectome organization.

We include, below, a figure summarizing the above analyses and document changes made to the text.

FIG. S17. Forcing symmetry of weights degrades model performance. (a) Scatterplot of asymmetries in weights inferred using the model proposed here with asymmetries in weights estimated directly from tract-tracing data in mouse connectome (asymmetry is measured as $W_{ij} - W_{ji}$). We explored the effect of (a)symmetry by fitting models in which we forced edge weights to be symmetric. We considered three symmetrizing routines given a pair of edges W_{ij} and W_{ji} . First, we set both edge weights equal to their mean: $W_{ji} = W_{ij} = \frac{W_{ij} + W_{ji}}{2}$. We denote the matrix estimated from this procedure as *av-sym*. The second and third approaches involved mirroring the upper and lower elements of the connectivity matrix. That is, we set both weights equal to either W_{ij} or W_{ji} . We refer to the corresponding matrices as *u-sym* and *l-sym*. We then evaluated model performance using these weights. In general, we found that model performance was best using the original asymmetric weights in both mouse (b) and human data (c). This is not surprising, as those weights were estimated to optimize model fit. (d) Next, we tracked model performance across subjects in the human data using individual level SC and similarly found that model performance is best when the original asymmetric weights are used for prediction. (e) Finally, we tracked model performance locally for each brain region and we found that brain regions in heteromodal systems were more negatively effected by forcing symmetry in model weights than brain regions in unimodal systems (visual and motor systems). In addition, in order to check that these local changes to model performance were consistent we tracked these measures across subjects and we found that per region model performance was similar across subjects (f). The previous analyses showed that three procedures for symmetrizing weights lead to decrements in performance. It is not surprising that this is the case, as the symmetrized weights were never fit to data, making the comparison against the asymmetric weights inappropriate. As a final analysis, we use gradient descent to train regression models that estimate new weights on edges but can either allow for asymmetric weights or force weights to be symmetric. We fit these models to the 95 participants from the Human Connectome Project dataset. (g) This plot shows the loss trajectory across training for enforced symmetry (red) and models where asymmetry was allowed (blue). In general, we find that symmetric edges lead to reduced performance. Mean trajectories are colored in bold. (h) Boxplots comparing the MSE loss of asymmetric (blue) and symmetric models (red) after training for 200 steps of gradient descent (95 subjects). Asymmetric models had significantly lower loss (paired-sample *t*-test; $p < 10^{-15}$).

We reference this new analysis and figure in two places in the text. In the main results section on “Asymmetries in connection weights”:

“Finally, we performed an additional analysis to probe the importance of the asymmetries in our model for model performance. We found that forcing the weights of our trained model to be symmetric degraded model performance for both group-based assessments in mice (Fig. S17b) and humans (Fig. S17c) as well as individual based models (Fig. S17d,g-i). Additionally, we found that the regions whose predictions were most effected by forcing symmetry in connection weights were heteromodal regions (Fig. S17e).”

...and in a new/edited discussion section on **“Interpreting signed and asymmetric edges in a macroscale connectome”**:

“Indeed, another analysis found that while forcing our model weights to be symmetric results in worse model performance, the predictions that are most negatively effected are regions in heteromodal systems (Fig. S17).”

In addition to this new analysis, we also edited our section on **“Interpreting signed edges in a macroscale connectome”** to include a discussion of the asymmetric weights in our model. See below for a full copy of the new/edited section on **“Interpreting signed and asymmetric edges in a macroscale connectome”**:

“Two of the unique features of the networks we construct here are that edge weights are signed and directed. How do we interpret these features? Are there possible neurophysiological explanations?”

At their core, the edge weights estimated here are statistical constructs. Specifically, they represent how well the past activity of node i 's connected neighbors explains i 's future activity. For a given node i , its in-weights must also be interpreted as a group; they are estimated simultaneously, with neighbors of i competing for the pool of unexplained variance. In a simple two node system, a negative weight from node i to j indicates that when the activity of i increases, the activity of j tends to decrease proportionally at the next time point. As you increase the number of connected nodes, negative weights should be interpreted in the context of the neighbors of the node whose activity they are predicting. Similarly, asymmetric weights arise from modeling future activity from the “perspective” of different nodes, such that node i has greater influence over node j 's model than the reverse. This interpretation in which the connection, $W_{j \rightarrow i}$, represents how much the history of j explains the current state of i is broadly in line with extant modeling frameworks for estimating effective connectivity [74].

Although it is tempting to ascribe “excitatory” and “inhibitory” labels to positive and negative edge weights, this terminology is typically reserved for cell-to-cell projections. In general, the neurochemical (e.g. glutamate/excitatory and GABA/inhibitory) contributions to the diffusion MRI and fMRI BOLD signal are not easily parsed [75]. However, we can speculate about possible underlying mechanisms that support signed edges in large-scale networks. One possible explanation involves feed-forward or feedback inhibition [76], whereby excitatory inter-regional connections cause inhibition in their target region either by directly exciting local inhibitory interneurons, or by exciting local interneurons indirectly through connecting pyramidal cells. Indeed, recent studies have suggested that the balance of glutamate/GABA underlies the antagonistic (anti-correlated) activity of large-scale brain systems [77].

Similarly, although it is tempting to ascribe a causal story to the asymmetric weights of our model such that one region has more causal influence over another region, causality is notoriously difficult to assess, especially in complex systems like the brain [78, 79]. Many methods have been developed in attempt to disambiguate causal asymmetries in

brains including direct electrical stimulation of brain regions [80] and effective connectivity and dynamic causal models [49–54]. Still, the difficulty involved in characterizing causality in complex systems warrants a multi-pronged approach. With the asymmetric, weighted and signed connectome, we hope to provide an additional, easily calculated metric by which to assess asymmetric relationships between brain regions. With multiple metrics we might be able to better triangulate on the reality of causal asymmetries in these systems.

Indeed, in a supplementary analysis we found that the asymmetric weights estimated using our model are uncorrelated with weights estimated from tract-tracing data (which offers an anatomical perspective on asymmetric connectivity) [1]. These observations suggest that incorporating functional information may yield largely complementary information about brain connectivity to traditional edge weight metrics. In support of this, a number of our findings suggest that the weights in our model are not randomly distributed throughout the brain, but reflect known organizational axes of brain networks. For instance, asymmetric edges with different signs (+/-) tend to fall between large-scale brain systems rather than within and the similarity of in-strength and out-strength is lowest in heteromodal poles, suggesting that asymmetry might be a hallmark of polyfunctionality.

Irrespective of their underlying origins, the signed and asymmetric edges in the networks constructed here exhibit non-random organization in terms of their distribution across canonical brain systems and relationship to other network/geometric measures—e.g. clustering coefficient and fiber length. These features, of course, have implications for traditional network analyses and may spur methodological innovation within the field of network neuroscience. For instance, interregional communication models rely on shortest paths and diffusion dynamics to estimate communication efficacy [23, 81]. However, these measures are not well-defined for networks with signed edges. Here, we circumvent this issue by offsetting edge weights, forcing negative connections to have small (but positive) values. However, there are likely many alternative strategies that embrace the signed nature of edges that could be explored in future studies [82].”

R2. Comment #3:

- One of the major conceptual limitations when estimating SC from dMRI is that dMRI-tractography follows water and not necessarily axons, hence these matrices can have many false positives. The proposed approach seems to ignore this fact and subsequently takes an SC matrix for granted, binarise it and learn its weights from a different modality. I think it would be great to get a feel of how false positive connections in the binarised graph are treated in the weighted graph. For instance, do known false positives get assigned small weights after regression with the fMRI data? Or are false negatives upscaled? Hence the weighting can be used to somehow “filter” the original matrix?

Or the approach simply propagates/amplifies false positives, in which case interpretation issues and challenges remain unresolved? One would hope that a false positive connection (that reflects dMRI limitations) won't be important in predicting fMRI data (that are independent to dMRI and

their limitations). Such type of indirect “filtering” of SC using multi-modal data would be a genuine contribution to the field and would be worth exploring with the proposed framework in my opinion. And would considerably support the claim about redefining SC.

- Along the same lines, in Fig. 1. explorations are presented on testing how stable the regressed weights are when subsampling the fMRI data. But what is not tested is how stable these weights are to perturbations of the original SC matrix. An SC matrix can be obtained in a number of ways, binarising it also can be done in a number of ways. How sensitive or not the results are to how this SC matrix is obtained? Particularly given that it contains errors?

Again, the review brings up an important point, in this case related to the verisimilitude of structural connections inferred from dMRI+tractography. We agree that it is important to assess what effect tractography errors may have on our model. To explore this, we conducted three additional analyses (see Fig. S16, Fig. S18, and Fig. S19).

First, to assess the impact of potential false positives and negatives, we conducted an additional analysis in which we treated the SC matrix as a “gold standard” and swapped (rewired) detected connections with connections that were initially absent from our SC matrix, gradually degrading the original matrix. We note that while it is certainly untrue that the SC matrix is error-free, we speculate that the proportion of false positives is far lower than the proportion of true negatives—i.e. a small fraction of regions are connected errantly, while the majority of disconnected regions are correctly disconnected. Therefore, iteratively replacing existing connections with connections that were initially missing will generate surrogate connectomes composed of far more false positives than the original. This method served as a proxy to assess how our model handles invalid connections, akin to addressing false positives.

Specifically, we started with the intact network. We then selected one of the M original edges and replaced it with another edge of approximately the same length (interregional Euclidean distance; see Betzel et al 2019, Network Neuroscience), thereby ensuring that the surrogate connectome has approximately the same “wiring cost” as the original. We repeated this swapping procedure until each of the original edges had been replaced.

In general, we found as the number of swaps increased (corresponding to a greater proportion of false positives), there was a noticeable degradation in model performance (see Fig. S16b,c). This suggests that our model is sensitive to the verisimilitude of connections in the SC matrix.

We also note that, despite the fact that we completely replaced every existing edge, the weights assigned to the swapped edges remained relatively consistent. More specifically, we saw that the absolute difference between the weights of these swapped edges was less than a standard deviation of the weights used in our main text. This indicates that our model does not disproportionately upscale false negatives or diminish false positives.

We also performed two additional analyses wherein we varied the method and parameters used to generate the group-level connectome. First, we varied the density parameter in the “distance-dependent” method used in the original submission (Betzel et al 2019). This approach is similar to other consistency-based approaches in that it seeks to retain connections that are consistently

expressed across the majority of participants, while excluding more variable connections. Unlike other consistency-based approaches, which can artifactually retain more short-range connections, this approach matches the edge length distribution of subject-level connectomes, maintaining balance in terms of the number of short-/middle-/long-distance connections it retains. It includes one free parameter: the total number of connections desired. In the original submission, we fixed this value to the mean number of connections across participants. Here, we explored different values for this parameter by scaling the mean number of connections to retain to a range from 0.17 to 1.82, arriving at connectomes with densities from 3.3% to 31.3% (5240 to 50008 edges; we note that this range is extreme when compared to the actual range of density values between individual subject connectomes; range = 15.6%-20%, or 24902 to 31460 edges). Second, and in the interest of completeness, we also performed the more standard consistency-based thresholding for construction of the group-representative matrix. We varied the consistency threshold so as to match the densities of the distance-based consistency method.

Using these alternative group-representative connectomes, we estimated edge weights using the procedure described in the main text. Our aim was to show that across these different connectomes the edge weight structure was largely preserved. This is not straightforward because, by design, the connectomes contained different sets of connections. Accordingly, for every pair of connectomes, we compared only the weights over the intersection of edges – i.e. edges expressed in *both* connectomes.

In general, we found that weights were highly similar across different estimates of the group-representative connectome; the lowest similarity values were $r = 0.7$ for the distance-dependent group connectome, and $r = 0.85$ for the more common consistency-based threshold; see Fig. S19e and S18e respectively). Additionally, we found that these edges were even more similar when the underlying structural connectomes maintained high Jaccard similarity (see Fig. S19i and S18i; correlation between edge-weight similarity and Jaccard similarity: $r = 0.9$, and $r = 0.47$ respectively).

We now include these results in the main text. See below for figures (Fig. S16, Fig. S18, and Fig. S19) highlighting results described above.

We referenced these figures in the main **Results** section entitled “**Fitting and benchmarking asymmetric, weighted, and signed structural connectivity**”:

“In addition, we performed several analyses to assess how our model is effected by changes to the underlying structural network. We found that as existing structural connections are replaced with non-existing connections (with equal distance) performance of the model degrades (Fig. S16). Furthermore, we found that, in general, changes to the underlying structural network -- for example by changing parameters in the estimation of group consensus structural connectivity -- result in changes to the weights of the model. More specifically, the similarity of model weights is positively related to the similarity of the structural network (Fig. S18 and Fig. S19). Finally, although the weights of these models do change with changes to the underlying structural network, the lower bound on weight similarity is still reasonably high (between $r = 0.7$ and $r = 0.85$; see Fig. S19e and S18e). ”

FIG. S16. Replacing inferred SC connections with non-existent edges of the same distance degrades model performance. (a) Schematic illustrating procedure for perturbing empirical connectome. Briefly, we identified an existing connection (shown in black) and removed it; replacing it with a new connection (shown in green) of approximately the same distance. (b) As we continued to increase the number of edges that we changed, the model performance degrades, as measured by the correlation between the predicted and observed time series. Here, we show results from 10 different runs (different colored lines). (c) As we continued to increase the number of edges that we changed, the model performance degrades (as measured by the mean squared error between the predicted and observed time series). Shown across 10 different iterations. (d) Plot showing that the weight differences remain very small between original SC edges in the intact model versus the models that contain swapped edges. (e) Plot showing that the weight differences between swapped distance-matched SC edges remain very small. Shown across 10 different iterations where swapped edges were randomly selected. Y axis represents the mean weight difference across all swapped edges. It is important to note that all of these changes were below 1 standard deviation of the weights for the model used in the main text.

FIG. S18. Variation in consistency-based group SC calculation: similar networks produce similar weights. (a) Group structural connectivity matrices. The first matrix is a consistency matrix that shows the percentage of subjects across whom different structural connections are consistently present. The next set of matrices show how you can form different group-based structural connectivity matrices by thresholding the consistency matrix. We ran our model on the structural matrices produced across ten thresholds (10%,20%,30%,40%,50%,60%,70%,80%,90%,100%). In the following analyses, we compare the asymmetric, and signed weights of our model across these structural connectomes. More specifically, we compare the weights of edges that are present across all thresholds. (b) Matrix of the absolute edge weight differences between each of these thresholded models. (c) Matrix of the number of non-overlapping edges between each of these thresholded SC matrices. (d) Matrix of the number of overlapping edges between each of these thresholded SC matrices. (e) Matrix of the Pearson correlation of weights between each of these thresholded models. (f) Matrix of the Jaccard similarity between each of these thresholded SC matrices. (g) Plot showing the negative relationship between the average absolute weight difference of edges and the number of overlapping edges in the thresholded SC. (h) Plot showing the positive relationship between the weight similarity of edges and the number of overlapping edges in the thresholded SC. (i) Plot showing the positive relationship between the weight similarity of edges and the Jaccard similarity of the thresholded SC. (j) Plot showing the positive relationship between the average absolute weight difference of edges and the number of overlapping edges in the thresholded SC. (k) Plot showing the negative relationship between the weight similarity of edges and the number of non-overlapping edges in the thresholded SC.

FIG. S19. Variation in distance-dependent group SC calculation: similar networks produce similar weights. (a) Group structural connectivity matrices produced using a distance-dependent group consensus method described in [30], but varying a scale factor. These scale factors determine the proportion of edges in different distance bins to match with the average subject. We ran our model on the structural matrices produced across ten different scaling factors (1.82, 1.45, 1.21, 1.05, 0.9, 0.79, 0.67, 0.56, 0.4, 0.17) to approximately match the densities in our previous consistency-based analysis. It is important to note that a scale factor of 1 produces the exact same model as the model used in the main text. In the following analyses, we compare the asymmetric, and signed weights of our model across these structural connectomes. More specifically, we compare the weights of edges that are present across all thresholds. (b) Matrix of the absolute edge weight differences between each of these scaled models. (c) Matrix of the number of non-overlapping edges between each of these scaled SC matrices. (d) Matrix of the number of overlapping edges between each of these scaled SC matrices. (e) Matrix of the Pearson correlation of weights between each of these scaled models. (f) Matrix of the Jaccard similarity between each of these scaled SC matrices. (g) Plot showing the negative relationship between the average absolute weight difference of edges and the number of overlapping edges in the scaled SC. (h) Plot showing the positive relationship between the weight similarity of edges and the number of overlapping edges in the scaled SC. (i) Plot showing the positive relationship between the weight similarity of edges and the Jaccard similarity of the scaled SC. (j) Plot showing the positive relationship between the average absolute weight difference of edges and the number of overlapping edges in the scaled SC. (k) Plot showing the negative relationship between the weight similarity of edges and the number of non-overlapping edges in the scaled SC.

Minor comments

- The paper is quite long. Some sections can be shortened by moving whole paragraphs to supplementary and summarise in a sentence or two in main text (e.g. page 5, paragraph around Figures S10, S11, offers extra evidence, but nothing new in terms of the main message of the section, so this can all go into supplement. Similarly for other sections. In Methods for instance there are very long sections describing data and processing that are publicly available, I would focus these sections on the extra processing you did.)

We thank the reviewer for their careful reading of our manuscript. To shorten the main text, we abbreviated a number of paragraphs describing supplemental analyses, and referenced the figures and methods section for more details.

First, in the main results section entitled “**Modular organization of the asymmetric, weighted, and signed connectome**” we deleted the following paragraphs:

For completeness, we also examined the multi-scale and hierarchical organization of communities, allowing for the resolution γ to vary. These results suggest that the asymmetric, signed, and weighted network exhibits community structure not limited to a single topological scale.

*Notably, the results reported here were obtained using modularity maximization and a well-established null model. We also explored an alternative “geographic” null model that has been used in network analysis of physical systems, e.g. granular materials (details of this model are described in **Materials and Methods**). Briefly, this null model preserves the binary network architecture exactly -- the same presence/absence of links as in the observed network -- but assigns a uniform weight across those edges. In general, we find that this null model generates results consistent with those described above, but also yields consensus communities that exhibit a striking correspondence to canonical brain systems.*

Additionally, we compared the detected modules to a recently aggregated set of “brain maps” -- annotations of brain regions that describe properties ranging from density of receptors to the relative expansion of brain areas across development and evolution. In general, we found evidence that the modules detected using the asymmetric, weighted, and signed network were more strongly enriched for these annotations compared to modules detected in the fiber density matrix. This observation was true both at the level of the entire partition, but also at the level of individual modules.

We replaced these paragraphs with two shorter paragraphs:

“In addition, we conducted a number of supplemental analyses to explore the modular structure of these networks in more detail, providing evidence that they exhibit hierarchical community structure (Fig. S9), and that the modules from our model were more strongly enriched for “brain map” [18] annotations describing properties ranging from density of receptors to the relative expansion of brain areas across development and evolution (Fig. S12). In addition, we introduce a new “geographic” null model for use with modularity maximization (Fig. S11 and Fig. S10).

*Finally, we also repeated several of the analyses from this and the previous section using mouse anatomical connectivity data made available by the Allen Brain Institute [14] and fMRI data acquired from a cohort of $N = 18$ anaesthetized mice (see Fig. S13 and **Materials and Methods** for more details).”*

Next, in the main results section entitled “**Fitting and benchmarking asymmetric, weighted, and signed structural connectivity**” we replaced the following text describing null models with a reference to methods section where the null models were already described in greater detail:

*Accordingly, we compared the observed model fitness against null distributions obtained under five distinct null models: 1) a minimally wired null model in which only the shortest (least-costly) connections are preserved (while preserving an equal number of connections), 2) a reordered null model in which nodes' orders were randomly permuted, 3) a “spin” model in which nodes' orders were randomly permuted but geometry preserved, 4) a topological null model in which nodes' degrees (number of connections and predictors) were preserved, but edge placement randomized, and 5) a temporal null model in which regional fMRI BOLD time series were circularly shifted independently for each region and scan (see **Materials and Methods** for details related to these null models).*

We replaced this text with the following shorter sentence and reference to the “**Materials and Methods**” section:

*“Accordingly, we compared the observed model fitness against null distributions obtained under five distinct null models [19] (see **Materials and Methods** for details related to these null models).”*

Next, we replaced the following paragraph in the preamble to our “**Discussion**” section:

Here, we explored a simple regression-based model for endowing reconstructed fiber tracts with directionality and a signed weight. Benchmarking this method on Human Connectome Project data, we found that the model fit observed data well, outperforming a suite of null models. The estimated weights were highly reliable even when fit using relatively few training samples and exhibited marked subject specificity. We next analyzed the resulting network using tools from network neuroscience. These analyses revealed communities that spanned cerebral hemispheres and mapped clearly onto known functional systems. Almost every edge in this network was involved in at least one shortest path. We also found evidence of asymmetric weights, network reconfiguration during naturalistic movie watching, and age-related differences. We note that unlike biophysical and dynamic causal models, our weighting scheme is not generative--i.e. it cannot be used to generate new synthetic data. It is, however, explanatory and represents a means of weighting fiber tracts that is distinct from those most frequently used in network neuroscience. Collectively, the proposed framework presents opportunities for multiple follow-up studies and applications in other neuroscience disciplines.

We replaced this with the following shorter paragraph:

“We note that our approach shares some features with existing frameworks--e.g. effective connectivity and dynamic causal models [1-6] --in that it returns directed connections. However, our weighting scheme is not generative--i.e. it cannot be used to generate new synthetic time series data. Additionally, our approach is not seeking to solve the inverse

problem--i.e. inferring structure from function. Rather, it is explanatory and represents a means of weighting already reconstructed fiber tracts. It is therefore distinct from extant approaches in network neuroscience, and presents opportunities for multiple follow-up studies and applications in other neuroscientific disciplines.”

Additionally, we decided that the following text in the discussion section entitled “**A network bridge between anatomical and functional connectivity**” had already been said elsewhere:

Here, we explore this alternative approach for tracking changes in structure-function correspondence in two contexts: comparing edge weights between rest and movie-watching conditions and identifying differences in edge weight across the human lifespan.

We find robust reconfiguration of edge weights during movie-watching. This observation is not new--a seemingly limitless number of studies have reported task- or state-induced changes in connection weights. In these studies, however, it is the functional connections whose weights change, making it difficult to assess which structural connection facilitate those changes. Our approach addresses this issue directly; changes in regression weights at each structural edge allow for the formulation of targeted hypotheses about the roles of specific fiber tracts in a given task.

We note that other studies have aimed to link structure and function through generative models of brain activity--e.g. variations of dynamic causal models (DCMs). Generally, these studies estimate parameters--including connection weights--for biophysical models. The models are generative in the sense that, when iterated, they yield synthetic activity time courses that capture select aspects of the empirical time courses--e.g. activation profiles. Like our model, DCMs can yield signed and asymmetric connection weights. However, computational complexity limits their scale to networks of approx. 100 nodes and they are typically not constrained by anatomical connectivity--i.e. effective connections can be inferred between pairs of nodes that are not materially linked by a fiber pathway. Despite the outward similarity of DCMs and other methods for estimating “effective connectivity”, our approach remains distinct in that it serves only to estimate new weights for structural connections rather than as generative model of brain activity.

Additionally, we removed the following text from the “**Limitations**” section:

However, new connections mean including additional explanatory terms in each regional multi-linear model and will, in general, lead to new estimates of edges' weights. That is, the regression coefficients will vary with additional observations (new data) or new nodes. Future studies should investigate this explicitly.

We also removed the following text from the “**Future directions**” section:

Additionally, while the edges we weight are structurally-defined and reflect best estimates of white-matter topology, we endow them with functionally relevant edge weights (derived from fMRI BOLD data). The multi-modal nature of this edge-weighting

scheme may, in actuality, situate our approach somewhere between functional and anatomical connectivity. That is, it achieves what resting-state FC sometimes is assumed to be; namely, a functionally informative measure of anatomically connectivity.

- Fig. 4b, colour code should start from zero, as absolute values are shown

We thank the reviewer for their attention to detail. We have updated the colorbar on the figure panel to reflect the proper range for positive values. Please see below for a copy of the edited figure:

- I would make clear early on whether by BOLD activity the authors refer to resting-state or task.

We appreciate this attention to detail.

We have added the following text to the first main results section entitled “**Fitting and benchmarking asymmetric, weighted, and signed structural connectivity**” in order to clarify that we used resting-state fMRI data:

“In this section, we report the results of the fitted model on resting state fMRI data from human subjects.”

References

1. K. Seth and G. M. Edelman, "Distinguishing causal interactions in neural populations," *Neural computation*, vol. 19, no. 4, pp. 910–933, 2007.
2. A. Roebroeck, E. Formisano, and R. Goebel, "Mapping directed influence over the brain using granger causality and fmri," *Neuroimage*, vol. 25, no. 1, pp. 230–242, 2005.
3. L. Novelli, P. Wollstadt, P. Mediano, M. Wibral, and J. T. Lizier, "Large-scale directed network inference with multivariate transfer entropy and hierarchical statistical testing," *Network Neuroscience*, vol. 3, no. 3, pp. 827–847, 2019.
4. S. Frässle, E. I. Lomakina, A. Razi, K. J. Friston, J. M. Buhmann, and K. E. Stephan, "Regression dcm for fmri," *Neuroimage*, vol. 155, pp. 406–421, 2017.
5. Razi, M. L. Seghier, Y. Zhou, et al., "Large-scale dcms for resting-state fmri," *Network Neuroscience*, vol. 1, no. 3, pp. 222–241, 2017.
6. L. Novelli, K. Friston, and A. Razi, "Spectral dynamic causal modelling: A didactic introduction and its relationship with functional connectivity," *arXiv preprint arXiv:2306.13429*, 2023.
7. K. J. Friston, "Functional and effective connectivity in neuroimaging: A synthesis," *Human brain mapping*, vol. 2, no. 1-2, pp. 56–78, 1994.
8. G. Buzsáki, "Feed-forward inhibition in the hippocampal formation," *Progress in neurobiology*, vol. 22, no. 2, pp. 131–153, 1984.
9. H. Gu, Y. Hu, X. Chen, Y. He, and Y. Yang, "Regional excitation-inhibition balance predicts default-mode network deactivation via functional connectivity," *Neuroimage*, vol. 185, pp. 388–397, 2019.
10. A. Wagner, "Causality in complex systems," *Biology and Philosophy*, vol. 14, pp. 83–101, 1999.
11. G. Ugolini, "Advances in viral transneuronal tracing," *Journal of neuroscience methods*, vol. 194, no. 1, pp. 2–20, 2010.
12. Rogers and K. T. Beier, "Can transsynaptic viral strategies be used to reveal functional aspects of neural circuitry?" *Journal of neuroscience methods*, vol. 348, p. 109 005, 2021.
13. C. Seguin, M. Jedynak, O. David, S. Mansour, O. Sporns, and A. Zalesky, "Communication dynamics in the human connectome shape the cortex-wide propagation of direct electrical stimulation," *Neuron*, vol. 111, no. 9, pp. 1391–1401, 2023.
14. S. W. Oh, J. A. Harris, L. Ng, et al., "A mesoscale connectome of the mouse brain," *Nature*, vol. 508, no. 7495, pp. 207–214, 2014.
15. A. Avena-Koenigsberger, B. Misic, and O. Sporns, "Communication dynamics in complex brain networks," *Nature Reviews Neuroscience*, vol. 19, no. 1, p. 17, 2018.
16. A. Fornito, A. Zalesky, and E. Bullmore, *Fundamentals of brain network analysis*. Academic Press, 2016.

Reviewer #1 (Remarks on code availability):

I have no further issues.

Reviewer #2 (Remarks to the Author):

I would like to thank the authors for the amount of work they put to revise the manuscript and address all my comments. This is a very nice paper, I only have a minor suggestion, but I do not need to see the paper again. The authors can feel free to do whatever they want with the suggestion below.

In the response to Reviewers, the authors make very clear what the aims are and how their approach differs from dMRI-based approaches. For instance, they mention:

- "Our primary aim is to reweight or otherwise "inform" structural connections with functional imaging data. Through this lens, our approach functionally annotates existing tracts, assigning them weights that reflect neural activity..."

- "We propose integrating functional imaging data into the estimation of structural edge weights. We present evidence that this approach yields weights that are largely uncorrelated with existing dMRI measures, such as FA, MD, streamline counts."

I found the above two points very clear and convincing and I would incorporate something along these lines explicitly in the Introduction. I would refrain from using statements such as "we present a ... model for estimating the weights of structural connections", which are confusing, while the above arguments are very clear.

Reviewer #2 (Remarks on code availability):

There is a reasonable effort making the code reusable, with instructions and a README on what functions do what and how to go about regenerating results.

Relevant data is also shared.